# An electrophysiological marker of arousal level in humans

**Janna D Lendner[1,2]\*, Randolph F Helfrich[3,4], Bryce A Mander[5], Luis Romundstad[6], Jack J Lin[7], Matthew P Walker[1,8], Pal G Larsson[9], Robert T Knight[1,8]**

[1]Helen Wills Neuroscience Institute, University of California, Berkeley, Berkeley, United States; [2]Department of Anesthesiology and Intensive Care Medicine, University Medical Center Tuebingen, Tuebingen, Germany; [3]Hertie-Institute for Clinical Brain Research, Tuebingen, Germany; [4]Department of Neurology and Epileptology, University Medical Center Tuebingen, Tuebingen, Germany; [5]Department of Psychiatry and Human Behavior, University of California, Irvine, Irvine, United States; [6]Department of Anesthesiology, University of Oslo Medical Center, Oslo, Norway; [7]Department of Neurology, University of California, Irvine, Irvine, United States; [8]Department of Psychology, University of California, Berkeley, Berkeley, United States; [9]Department of Neurosurgery, University of Oslo Medical Center, Oslo, Norway

**Abstract** Deep non-rapid eye movement sleep (NREM) and general anesthesia with propofol are prominent states of reduced arousal linked to the occurrence of synchronized oscillations in the electroencephalogram (EEG). Although rapid eye movement (REM) sleep is also associated with diminished arousal levels, it is characterized by a desynchronized, 'wake-like' EEG. This observation implies that reduced arousal states are not necessarily only defined by synchronous oscillatory activity. Using intracranial and surface EEG recordings in four independent data sets, we demonstrate that the 1/f spectral slope of the electrophysiological power spectrum, which reflects the non-oscillatory, scale-free component of neural activity, delineates wakefulness from propofol anesthesia, NREM and REM sleep. Critically, the spectral slope discriminates wakefulness from REM sleep solely based on the neurophysiological brain state. Taken together, our findings describe a common electrophysiological marker that tracks states of reduced arousal, including different sleep stages as well as anesthesia in humans.

**\*For correspondence:**
janna.lendner@gmail.com

**Competing interests:** The authors declare that no competing interests exist.

## Introduction

General anesthesia is a reversible, pharmaceutically induced state of unconsciousness, while sleep is internally generated and cycles between rapid (REM) and non-rapid eye movement sleep (NREM; *Brown et al., 2010*; *Franks and Zecharia, 2011*). Both sleep stages and anesthesia are characterized by a behaviorally similar state of reduced physical arousal (*Brown et al., 2010*; *Franks and Zecharia, 2011*; *Murphy et al., 2011*). Definitions of arousal vary and include e.g. autonomic, behavioral or mental arousal. For this study, we followed an updated version of the framework by Laureys et al. that defined consciousness on two axes – *content* (awareness) and *level* (arousal; *Boly et al., 2013*; *Laureys, 2005*). While the conscious *content* is low in NREM sleep and propofol anesthesia, it is high in wakefulness and dreaming states like REM. The *arousal level* is low during anesthesia and in all sleep states including REM.

Both NREM sleep stage 3 (also called slow-wave sleep) and general anesthesia with propofol exhibit similar electrophysiological features, such as an increase in low frequency activity and the occurrence of prominent slow oscillations (<1.25 Hz; *Brown et al., 2010*; *Franks and Zecharia,*

**eLife digest** Electroencephalogram (EEG for short) is a widespread technique that helps to monitor the electrical activity of the brain. In particular, it can be used to examine, recognize and compare different states of brain consciousness such as sleep, wakefulness or general anesthesia. Yet, during rapid eye movement sleep (the sleep phase in which dreaming occurs), the electrical activity of the brain is similar to the one recorded during wakefulness, making it difficult to distinguish these states based on EEG alone.

EEG records brain activity in the shape of rhythmic waves whose frequency, shape and amplitude vary depending on the state of consciousness. In the EEG signal from the human brain, the higher frequency waves are weaker than the low-frequency waves: a measure known as spectral slope reflects the degree of this difference in the signal strength. Previous research suggests that spectral slope can be used to distinguish wakefulness from anesthesia and non-REM sleep. Here, Lendner et al. explored whether certain elements of the spectral slope could also discern wakefulness from all states of reduced arousal.

EEG readings were taken from patients and volunteers who were awake, asleep or under anesthesia, using electrodes placed either on the scalp or into the brain. Lendner et al. found that the spectral slope could distinguish wakefulness from anesthesia, deep non-REM and REM sleep. The changes in the spectral slope during sleep could accurately track the degree of arousal with great temporal precision and across a wide range of time scales.

This method means that states of consciousness can be spotted just from a scalp EEG. In the future, this approach could be embedded into the techniques used for monitoring sleep or anesthesia during operations; it could also be harnessed to monitor other low-response states, such as comas.

2011; Murphy et al., 2011; Prerau et al., 2017; Purdon et al., 2013). Moreover, propofol anesthesia has been linked to the emergence of a strong frontal alpha oscillation (8–12 Hz; Purdon et al., 2013) whereas spindles (12–16 Hz) typically appear in NREM sleep stage 2 (Prerau et al., 2017). In contrast, REM sleep is characterized by a desynchronized, active pattern in the electroencephalogram (EEG), which resembles wakefulness (Brown et al., 2010; Prerau et al., 2017). The additional defining features of REM sleep are therefore peripheral markers including muscle atonia as detected by electromyography (EMG) combined with rapid eye movements in the electrooculogram (EOG; Prerau et al., 2017). To date, it has been challenging to differentiate REM sleep from wakefulness in humans solely from the electrophysiological brain state (Pal et al., 2016).

Recently, several lines of inquiry highlighted the importance of non-oscillatory, scale-free neural activity for brain physiology and behavior (Miller et al., 2009a; Gao et al., 2017; Voytek et al., 2015; Voytek and Knight, 2015; Miller et al., 2009b; He et al., 2010). The electrophysiological power spectrum is characterized by a 1/f signal drop-off, i.e. higher frequency activity exhibits reduced power as compared to low frequency activity. This scaling law between power and frequency can be estimated from the exponential decay of the power spectrum (He et al., 2010) and has previously been used to assess a variety of cognitive and EEG phenomena (Colombo et al., 2019; Lina et al., 2019; Miskovic et al., 2019; Pereda et al., 1998; Pritchard, 1992; Shen et al., 2003; Susmáková and Krakovská, 2008). Notably, this decay function mainly captures non-oscillatory brain activity, which is not characterized by a defining temporal scale, such as band-limited oscillations (He et al., 2010). Therefore, analyses of scale-free 1/f dynamics might prove especially helpful when analyzing brain states that are not characterized by prominent oscillations such as REM sleep in humans. We hypothesized that markers of 1/f activity, such as the spectral slope of the power spectrum, may provide an electrophysiological signature that distinguishes 'paradoxical' REM sleep (Siegel, 2011) from wakefulness.

Importantly, 1/f dynamics can also be observed in a variety of other signals. For instance, long-range temporal correlations of neuronal oscillations (Linkenkaer-Hansen et al., 2001) or the size and duration of neuronal avalanches (Beggs and Plenz, 2003; Palva et al., 2013) also follow a power law but these scale-free behaviors likely have a different neurophysiological basis than the 1/f drop-off of the power spectrum (He et al., 2010).

Recent findings suggested that 1/f dynamics differentiate wakefulness from general anesthesia (*Colombo et al., 2019*; *Gao, 2016*). For instance, using intracranial recordings in macaque monkeys, it had been shown that the spectral slope between 30 and 50 Hz reliably tracked changes in arousal level under propofol anesthesia from induction to emergence (*Gao et al., 2017*). Moreover, it has been reported that that the spectral slope between 1 and 40 Hz in human scalp EEG recordings discriminated states with conscious content, namely wakefulness and ketamine anesthesia, from states where no conscious report was possible, i.e. Xenon and propofol anesthesia (*Colombo et al., 2019*). Collectively, these studies implied that propofol anesthesia was accompanied by a steeper decay of the power spectrum (*Colombo et al., 2019*; *Gao et al., 2017*).

With regard to sleep physiology, it had been observed that the spectral exponent of human scalp EEG becomes more negative during NREM sleep, when estimated e.g. in the 1 to 5 Hz (*Shen et al., 2003*), 3 to 30 Hz (*Pereda et al., 1998*) or 0.5 to 35 Hz frequency range (*Miskovic et al., 2019*). A similar pattern was observed in intracranial recordings with subdural grid electrodes in humans between 10 and 100 Hz (one subject; *Freeman and Zhai, 2009*) or 1 and 100 Hz (five subjects; *He et al., 2010*). Note that the 1/f background activity was estimated from frequency bands that were potentially influenced by simultaneously occurring low frequency oscillation i.e. delta (<4 Hz) or slow waves (<1.25 Hz) that might affect the degree of spectral tilt.

General anesthetics like propofol, etomidate and barbiturates act on GABAergic receptors to enhance inhibition (*Brown et al., 2011*). Recently, computational simulations indicated that the spectral slope might provide a surrogate marker for the excitation to inhibition (E/I) balance with more negative spectral slopes (esp. in the 30 to 50 Hz range) indexing enhanced inhibition (*Gao et al., 2017*). This model was validated using intracranial recordings in macaques and rodents: A shift in E/I-balance towards inhibition by administrating propofol resulted in a steeper slope of the power spectrum. Likewise, the spectral slope in the rodent hippocampus varied across the depth of hippocampus, directly reflecting the ratio of excitatory to inhibitory cells in the underlying neuronal population. Moreover, a modulation of spectral slope was also observed as a function of the hippocampal theta cycle, likely reflecting rapid shifts in E/I-balance (*Gao et al., 2017*).

A recent study that employed two-photon calcium imaging in mice provided additional insight into putative changes in E/I-balance during the sleep cycle. Cortical activity in mice was reduced during NREM sleep compared to wakefulness and, notably, even further reduced during REM sleep (*Niethard et al., 2016*). Crucially, the authors observed a selective increase in inhibitory interneuron activity (parvalbumin-positive interneurons; *Niethard et al., 2016*) during REM but not NREM sleep revealing an overall shift towards inhibition during REM sleep.

In the present study, we assessed if 1/f spectral dynamics, in particular in the 30 to 50 Hz range, which is devoid of prominent low-frequency oscillatory activity (*Gao et al., 2017*), could track arousal states in humans both under anesthesia with propofol and during sleep in intracranial and scalp EEG recordings. Specifically, we hypothesized that the spectral slope should become more negative (i.e. the power spectrum steeper) in sleep and under anesthesia compared to wakefulness. Importantly, we also predicted that the spectral slope could discriminate wakefulness from NREM as well as REM sleep. Based on recent reports linking E/I-balance and electrophysiology (*Gao et al., 2017*; *Niethard et al., 2016*), we reasoned that the spectral slope, as a putative marker of E/I-balance, should facilitate the detection of REM sleep directly from the current brain state, without complementary information from additional EMG or EOG recordings. While previous studies that included lower frequency power in their slope estimates, found the slope of REM to be close to wakefulness (*He et al., 2010*), we specifically investigated if the aperiodic background activity in the 30 to 50 Hz range could reliably discriminate REM sleep from wakefulness and NREM sleep.

## Results

We tested if non-oscillatory brain activity as quantified by the spectral slope of the electrophysiological power spectrum could discriminate different states of arousal in four independent studies: We obtained both (1) scalp EEG (n = 9) and (2) intracranial EEG (n = 12) under general anesthesia with propofol. Furthermore, we recorded (3) scalp EEG (n = 20) as well as (4) scalp EEG combined with intracranial EEG (n = 10) during a full night of sleep. We utilized both extra- and intracranial recordings to assess the precise spatial extent of the observed effects. In line with previous reports, the spectral slope was defined by a linear fit to the power spectrum in log-log space between 30 and 45

Hz (*Gao et al., 2017*). Critically, we carefully validated the chosen parameters in a series of control analyses as indicated below.

## 1/f spectral dynamics during propofol anesthesia

We first tested if the spectral slope discriminates wakefulness and propofol anesthesia in humans in two experiments. In the first study, we recorded scalp EEG during general anesthesia for orthopedic surgery in otherwise healthy adults (Study 1, n = 9). In the second study, we obtained intracranial EEG in epilepsy patients who were implanted with intracranial electrodes for seizure onset localization while they underwent general anesthesia for electrode explantation (Study 2, n = 12; subdural grid electrodes (electrocorticography; ECoG) and stereotactically placed depth electrodes (SEEG; coverage see *Figure 1—figure supplement 1a*).

In Study 1 (n = 9), we found that the time-resolved spectral slope closely tracked changes in arousal levels while subjects underwent propofol anesthesia (*Figure 1a*). Specifically, we observed a significant decrease of the spectral slope from wakefulness (−1.84 ± 0.30; mean ± SEM) to anesthesia (−3.10 ± 0.20) when averaged across all electrodes (*Figure 1b*; *permutation t-test*: p<0.0001, obs. $t_8$ = 7.09, $d_{Wake-Anesthesia}$ = 1.65).

A cluster-based permutation test assessing the spatial extent of this effect on the scalp level resulted in a single large cluster that spanned all 25 electrodes without a clear peak (p<0.001; *Figure 1b*). To further examine the spatial distribution of the observed scalp EEG pattern and to assess subcortical contributions, we utilized intracranial recordings in Study 2 (n = 12). Again, we

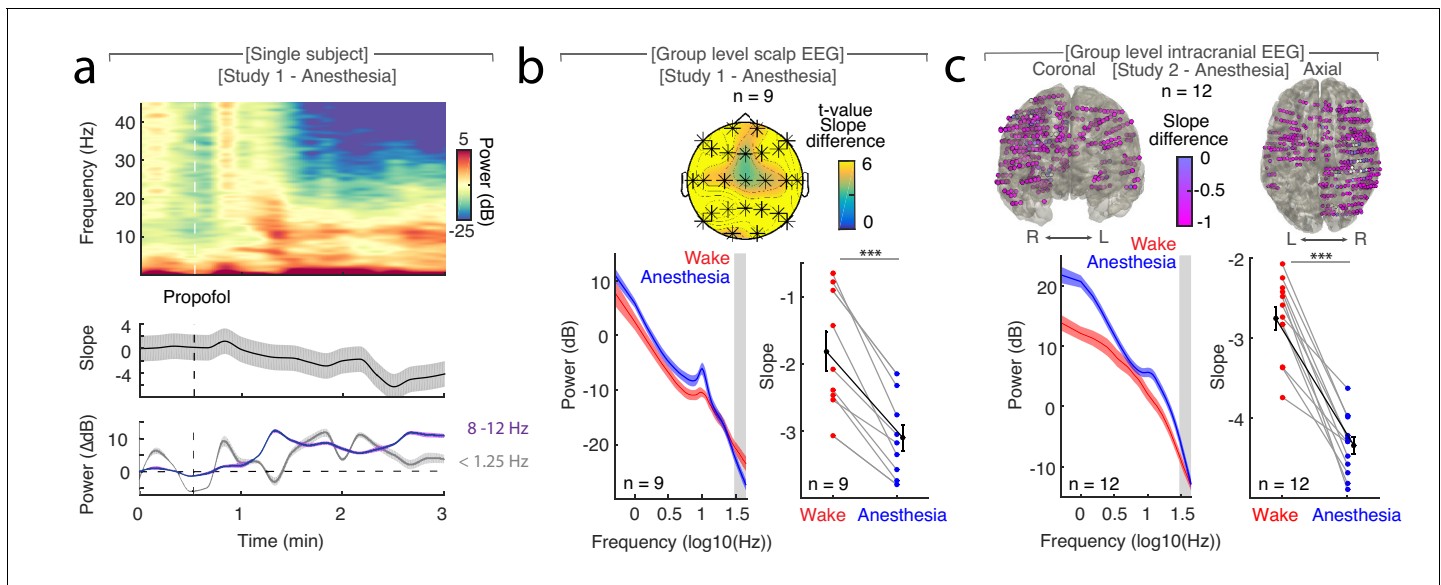

**Figure 1.** The spectral slope tracks changes in arousal level under general anesthesia with propofol. (**a**) Time-resolved average of three frontal EEG channels (F3, Fz, F4) during anesthesia. Upper panel: Time-frequency decomposition. Dotted white line: Induction with propofol. Middle: Spectral slope (black; mean ± SEM). Lower panel: Slow frequency (<1.25 Hz; gray) and alpha (8–12 Hz; purple) baseline-corrected power (mean ± SEM). Note, elevated slow frequency activity is already present during wakefulness. While alpha frequency activity is steadily increasing in the first minutes of anesthesia, slow frequency activity exhibits a waxing and waning pattern which may reflect the premedication with a sedative. (**b**) Anesthesia in scalp EEG (n = 9). Upper panel: Spatial extent of spectral slope difference. *Cluster permutation test*: *p<0.05. Lower panel: Left - Power spectra (mean ± SEM); Right – Spectral slope. Wakefulness (red), anesthesia (blue) and grand average (black; all mean ± SEM). *Permutation t-test*: ***p<0.001. (**c**) Anesthesia in intracranial recordings (n = 12). Upper panel: Left – coronal, right – axial view of intracranial channels that followed (magenta) or did not follow (white) the EEG pattern of a lower slope during anesthesia compared to wakefulness. Lower panel: Left – Power spectra; Right – Spectral slope. Wakefulness (red), anesthesia (blue) and grand average (black; mean ± SEM). *Permutation t-test*: ***p<0.001.

The online version of this article includes the following figure supplement(s) for figure 1:

**Figure supplement 1.** Coverage in intracranial subjects.
**Figure supplement 2.** Differences in spectral slope under general anesthesia and in sleep in cortical recording sites.
**Figure supplement 3.** The influence of segment length and number of tapers on spectral slope estimation under general anesthesia with propofol.
**Figure supplement 4.** The influence of different fit algorithms and fit lengths on spectral slope estimation in general anesthesia with propofol.

observed that the spectral slope was higher during wakefulness ($-2.75 \pm 0.15$) than during anesthesia ($-4.34 \pm 0.11$) when averaged across all electrodes (*Figure 1c*; *permutation t-test:* p<0.0001, obs. $t_{11} = 9.93$, $d_{Wake-Anesthesia} = 3.57$). This effect was present at the majority of recording sites (470 of 485 SEEG (96.9%); *Figure 1c*, *Table 1*). Notably, recordings from subdural grid electrodes (n = 4) showed the same pattern: The spectral slope decreased from wakefulness to anesthesia in the majority of recording sites (129 of 147 ECoG (87.75%); *Figure 1—figure supplement 2a*).

Taken together, we observed a more negative spectral slope under anesthesia compared to wakefulness in both scalp as well as intracranial EEG (*Figure 1b,c*). Our results indicate that the spectral slope differentiates between wakefulness and general anesthesia in humans. This effect spanned all scalp and the majority of intracranial electrodes, hence, supporting the notion that propofol anesthesia induces a global, brain-wide state change (*Brown et al., 2010*).

## 1/f spectral dynamics discriminate wakefulness, NREM and REM sleep

Having established that the spectral slope differs significantly between wakefulness and propofol anesthesia, we next examined if this state-dependent modulation generalized to other forms of decreased arousal, such as sleep. We specifically sought to determine if the spectral slope could discern wakefulness from different sleep stages. We analyzed two datasets obtained during a full night of sleep. In Study 3, we obtained polysomnography recordings from 20 healthy subjects, which included scalp EEG, as well as electrocardiography (ECG), electromyography (EMG) and electrooculography (EOG). To determine the precise spatial extent and subcortical contributions, we again recorded intracranial EEG in a separate cohort for Study 4 (n = 10; electrode coverage see *Figure 1—figure supplement 1b*). Critically, we combined intracranial EEG with polysomnography (scalp EEG, ECG, EMG, EOG) to enable comparable sleep staging across both the scalp and intracranial studies.

We observed that the time-resolved spectral slope closely tracked the technician-scored hypnogram (*Figure 2a*). To quantify this effect, we compared spectral slope estimates across wakefulness, N3 and REM sleep. In Study 3, we obtained a separate baseline eyes-closed recording during rest in 14 out of 20 subjects. In this subset, we observed prominent slope differences between quiescent rest ($-1.87 \pm 0.18$; mean $\pm$ SEM), N3 sleep ($-3.46 \pm 0.16$) and REM sleep ($-4.73 \pm 0.23$; *Figure 2b*). These differences were significant when averaged across all scalp EEG channels (*repeated-measures ANOVA permutation test:* p<0.0001, obs. $F_{1.94,\ 25.17} = 56.05$, $d_{Rest-Sleep} = 3.07$). Notably, N2 sleep exhibited an average slope of $-3.67 \pm 0.10$ that was also significantly below rest (*Figure 2—figure supplement 1a*; *permutation t-test:* $p_{Rest-N2}$ <0.0001; obs. $t_{13} = 7.97$; $d_{Rest-N2} = 3.31$). Permutation t-tests revealed a significant difference between rest and N3 ($p_{Rest-N3}$ <0.0001, obs. $t_{13} = 5.69$, $d_{Rest-}$

**Table 1.** Anatomical distribution of stereotactically placed intracranial depth electrodes in Study 2 – Intracranial anesthesia (n = 12).

| Brain region | Total number of electrodes | Electrodes with state-dependent slope modulation |
|---|---|---|
| ALL | 485 | 470 (96.9 %) |
| Prefrontal Cortex (PFC) | 179 | 175 (97.8 %) |
| medial Prefrontal Cortex (mPFC) | 27 | 27 (100 %) |
| lateral Prefrontal Cortex (lPFC) | 147 | 143 (97.3 %) |
| Orbito-frontal Cortex (OFC) | 5 | 5 (100 %) |
| Medial temporal Lobe (MTL) | 40 | 1 (95.0 %) |
| Hippocampus | 26 | 24 (92.3 %) |
| Amygdala | 13 | 13 (100 %) |
| Cingulate Cortex | 22 | 22 (100 %) |
| Insula | 13 | 13 (100 %) |
| M1/Premotor | 48 | 47 (97.9 %) |
| Lateral Temporal Cortex (LTC) | 50 | 50 (100 %) |
| Parietal Cortex | 84 | 78 (92.9 %) |
| Visual Cortex | 49 | 47 (95.9 %) |

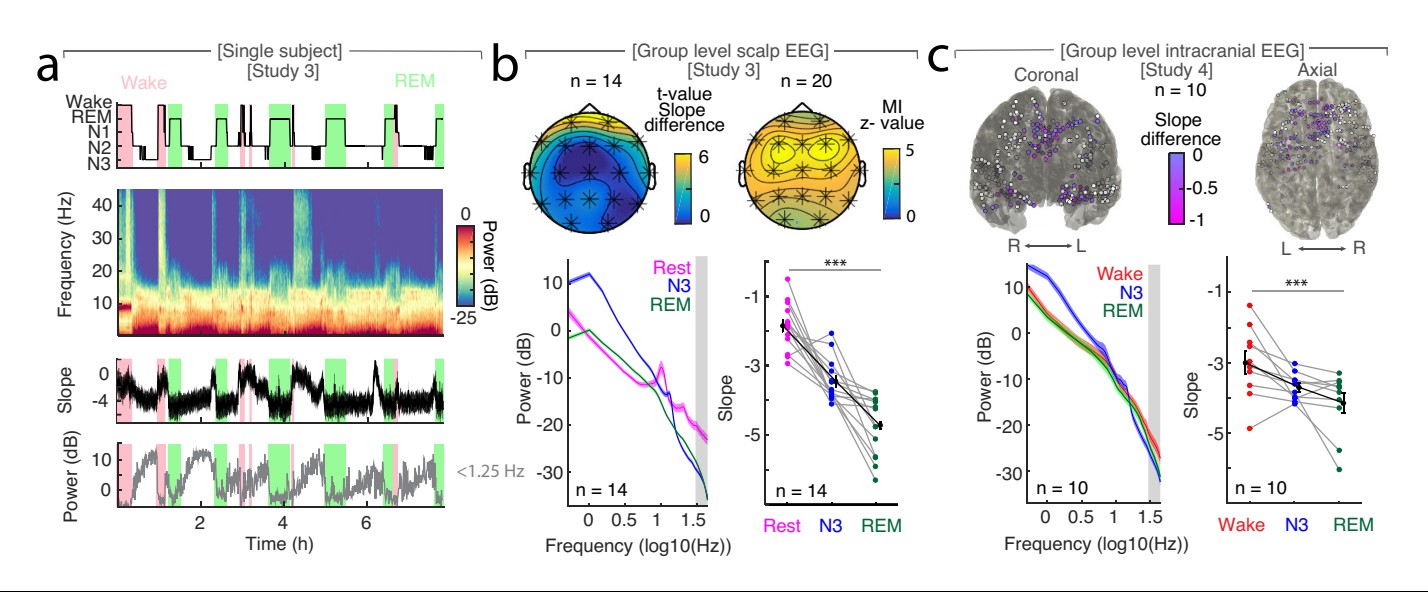

**Figure 2.** The spectral slope tracks changes of arousal level in sleep. (a) Time-resolved average of three frontal EEG channels (F3, Fz, F4) during a night of sleep. Upper panel: Expert-scored hypnogram (black), wake (pink), REM (light green). Upper middle: Time-frequency decomposition. Lower middle: Spectral slope (black; mean ± SEM). Lower panel: Slow frequency (<1.25 Hz) power (gray; mean ± SEM). (b) Sleep in scalp EEG. Upper panel: Left: Slope difference between sleep and rest (n = 14). *Cluster permutation test:* *p<0.05. Right: Mutual Information (MI) between the time-resolved slope and hypnogram (n = 20). *Cluster permutation test against surrogate distribution created by random block swapping:* *p<0.05. Lower panel: Left - Power spectra (n = 14; mean ± SEM); Right – Spectral slope (n = 14). Rest (magenta), NREM stage 3 (blue), REM sleep (green) and grand average (black; mean ± SEM). *Repeated measures ANOVA permutation test:* ***p<0.001. (c) Sleep in intracranial EEG (n = 10). Upper panel: Left – coronal, right – axial view of intracranial channels that followed (magenta) or did not follow (white) the EEG pattern of a lower slope during sleep (REM/N3). Lower panel: Left – Power spectra (mean ± SEM); Right – Spectral slope of simultaneous EEG recordings (Fz, Cz, C3, C4, Oz). Wakefulness (red), NREM stage 3 (N3; blue), REM sleep (green) and grand average (black; mean ± SEM). *Repeated measures ANOVA permutation test:* ***p<0.001.

The online version of this article includes the following figure supplement(s) for figure 2:

**Figure supplement 1.** Relative changes of spectral slope reliably differentiate between wakefulness, sleep and general anesthesia.

**Figure supplement 2.** The spectral slope is not confounded by muscle activity.

**Figure supplement 3.** Anatomical distribution of sleep dependent spectral slope modulation in scalp EEG.

**Figure supplement 4.** Differences of spectral slope in intracranial electrodes between waking and NREM sleep stage three or REM sleep (n = 10).

**Figure supplement 5.** Evaluation of different slope fit settings in intracranial sleep.

**Figure supplement 6.** The influence of different power calculations on signal-to-noise ratio during sleep.

**Figure supplement 7.** The influence of segment length and number of tapers on spectral slope estimation in sleep.

**Figure supplement 8.** The influence of reference schemes on spectral slope estimation during sleep.

**Figure supplement 9.** The influence of different fitting algorithms on spectral slope estimation during sleep.

**Figure supplement 10.** Comparison of Mutual Information captured by fronto-parietal connectivity and spectral slope.

**Figure supplement 11.** Slope difference between N3 and REM sleep.

$_{N3}$ = 2.49), between rest and REM ($p_{Rest-REM}$ <0.0001, obs. $t_{13}$ = 11.67, $d_{Rest-REM}$ = 3.71) and between N3 and REM sleep ($p_{N3-REM}$ = 0.0001, obs. $t_{13}$ = 4.44, $d_{N3-REM}$ = 1.70). Importantly, while some overlap of absolute spectral slope values between rest and sleep existed when comparing across individuals (*Figure 2—figure supplement 1a*), we observed a consistent individual decrease of – 2.06 ± 0.21 (mean ± SEM) between rest and all sleep stages (*Figure 2—figure supplement 1b*; Rest-N1 = −1.95 ± 0.26, Rest-N2 = −1.81 ± 0.23, Rest-N3 = −1.59 ± 0.28, Rest-REM = −2.86 ± 0.25).

Including all available wake periods (before, during and after the night of sleep in all 20 subjects) increased the variance (*Figure 2—figure supplement 1c*), which can be explained by the fact that subjects were still drowsy and data during state transitions was included. However, the overall pattern was remarkably similar (*Figure 2—figure supplement 1b,d*). As this approach increased the available amount of data, we utilized all wake trials (referred to as wake) for subsequent analyses.

Next, we assessed the spatial topography of where the slope tracks the hypnogram. Thus, we calculated the Mutual Information (MI) between the time-resolved spectral slope and the hypnogram.

MI is ideal to assess the relationship between a discrete variable (hypnogram) and neurophysiologic data (*Quian Quiroga and Panzeri, 2009*). In addition, we also repeated all analyses based on linear rank correlations, which yielded comparable results (*Figure 2—figure supplement 2a,b*).

We observed that the spectral slope closely tracked the hypnogram at all electrodes as indicated by a permutation test (n = 20; average z score = 4.14 ± 0.41 (mean ± SEM); all z > 2.8 correspond to a Bonferroni-corrected p<0.01; *Figure 2b*). This effect peaked over frontal electrodes F3, Fz and F4 (z = 4.90 ± 0.37; *Figure 2b*). Since frequencies of cranial muscle activity overlap with the frequency range used for spectral slope estimation, we controlled for possible muscle artifacts by repeating the analysis after local referencing (Laplacian; *Fitzgibbon et al., 2013*). In addition, we utilized partial correlations that considered the slope of the EMG as a confounding variable. These control analyses indicated that excluding these confounds strengthened the observed relationship between the hypnogram and the spectral slope (*Laplacian*: $p_{Spearman}$ <0.001, $p_{MI}$ <0.0001; *partial correlation*: $p_{Spearman}$ <0.001; *Figure 2—figure supplement 2*).

## Spatial characteristics of sleep state-dependent spectral slope modulations

We established that the spectral slope closely tracks the hypnogram. However, we observed pronounced differences between scalp electrodes (*Figure 2b*), thus, raising the question, which brain regions contribute most to the observed effects at the scalp level. In a source level analysis using an LCMV beamformer, prefrontal areas exhibited the strongest sleepstate-dependent modulation (*Figure 2—figure supplement 3*). To further investigate the contribution of cortical and subcortical regions, we obtained intracranial EEG recordings (Study 4, n = 10), which were combined with simultaneous scalp EEG recordings.

First, we aimed to replicate the results from Study 3. Again, we found that the slope decreased from wakefulness (−2.99 ± 0.32; mean ± SEM) to N3 sleep (−3.69 ± 0.12) and REM sleep (−4.15 ± 0.29; *Figure 2c*). These three states were significantly different in a repeated-measures ANOVA permutation test (p=0.0009; obs. $F_{1.97,\ 17.74}$ = 10.79, $d_{Wake-Sleep}$ = 1.12), thus, directly replicating the pattern as observed in Study 3. Permutation t-tests revealed a significant difference between wakefulness and REM (p=0.0002; obs. $t_9$ = 4.78; d = 1.19) and wakefulness and N3 (p=0.0136; obs. $t_9$ = 2.66; d = 0.92) but only marginally between N3 and REM (p=0.0431; obs. $t_9$ = 1.84; d = 0.64).

Second, we directly tested which intracranial SEEG contacts mirrored the observed scalp EEG pattern. We observed the same pattern - a more negative spectral slope in N3 and REM sleep as compared to wakefulness - in 155 of 352 SEEG (44.03%; *chi-squared test against chance-level (33%)*: $X^2$ = 8.20, p=0.0042; *Figure 2c*, *Figure 2—figure supplement 3*). Importantly, this analysis revealed that medial prefrontal cortex (mPFC) and medial temporal lobe structures (MTL; details see *Table 2*, *Figure 2—figure supplement 3*) exhibit a significantly larger fraction of electrodes showing sleep state-dependent slope modulation compared to their lateral counterparts (*chi-squared tests:* mPFC - lateral PFC: *p<0.0001, $X^2$ = 33.56*, MTL – lateral temporal cortex: p<0.0001, $X^2$ = 33.12), hence, converging on the same brain regions known to be the most relevant for sleep-dependent memory consolidation (*Dang-Vu et al., 2008*; *Helfrich et al., 2018*; *Mander et al., 2013*; *Murphy et al., 2009*).

Note that we did not specifically target any brain regions and in contrast to previous studies using subdural grid electrodes (*Gao et al., 2017*; *He et al., 2010*), the majority of our probes were stereotactically placed depth electrodes (for Wake - N3 and Wake - REM see *Figure 2—figure supplement 4*; subdural grid electrodes see *Figure 1—figure supplement 2b,c*). Given the spatial heterogeneity of intracranial responses (*Parvizi and Kastner, 2018*), there was a remarkable convergence on medial PFC that resembled the pattern observed at source level (*Figure 2—figure supplement 3*) and the overlying scalp EEG electrode Fz (*Figure 2*).

## The spectral slope discriminates wakefulness from states of reduced arousal

Our findings provide evidence that the spectral slope reliably discriminates wakefulness from sleep. Multiple prior reports indicated that slow waves are a hallmark of decreased arousal states (*Brown et al., 2010*; *Franks and Zecharia, 2011*; *Murphy et al., 2011*). We directly compared how well slow wave activity and spectral slopes estimates differentiate arousal states using both a linear

**Table 2.** Anatomical distribution of stereotactically placed intracranial depth electrodes in Study 4 – Intracranial sleep (n = 10).

| Brain region | Total number of electrodes | Electrodes with state-dependent slope modulation |
|---|---|---|
| ALL | 352 | 155 (44.0 %) |
| Prefrontal Cortex (PFC) | 132 | 49 (37.1 %) |
| medial Prefrontal Cortex (mPFC) | 28 | 24 (85.7 %) |
| lateral Prefrontal Cortex (lPFC) | 73 | 15 (20.6 %) |
| Orbito-frontal Cortex (OFC) | 30 | 10 (33.3 %) |
| Medial Temporal Lobe (MTL) | 48 | 33 (68.8 %) |
| Hippocampus | 27 | 19 (70.4 %) |
| Amygdala | 18 | 14 (77.8 %) |
| Cingulate Cortex | 40 | 31 (77.5 %) |
| Insula | 41 | 21 (51.2 %) |
| M1/Premotor | 7 | 7 (100 %) |
| Lateral Temporal Cortex (LTC) | 79 | 13 (16.5 %) |
| Parietal Cortex | 0 | 0 |
| Visual Cortex | 3 | 1 (33.3 %) |
| Other | 2 | 0 |

discriminant analysis (LDA) and a multivariate general linear Model (GLM) to discriminate different sleep states based on either the spectral slope or slow wave activity in 18 subjects (two subjects had to be excluded due to insufficient wake trials).

Note that both the LDA classifier and the GLM were trained on the same values that were used in the univariate testing. Both the LDA classifier as well as the GLM output provide a quantitative metric, namely the accuracy of correctly classified trials for LDA and the unique explained variance quantified by eta squared for the GLM enabling a direct comparison between different conditions. The GLM offers the additional advantage of facilitating the assessment of the multivariate interaction of the spectral slope and slow wave power. Data were z-scored before modeling with GLM and LDA outputs were logit-transformed before comparison.

## Linear Discriminant Analysis

First, we directly tested if the spectral slope is superior in discriminating REM sleep from wakefulness. We found that classifier performance was enhanced using the spectral slope compared to slow wave power (*spectral slope*: 76.31 ± 3.61% (mean ± SEM), *slow wave power*: 61.50 ± 1.93%; *permutation t-test*: p<0.001, obs. $t_{17}$ = 3.73, d = 1.25; *Figure 3a*). This finding indicates that the spectral slope constitutes a marker that successfully discriminates REM sleep from wakefulness solely from the electrophysiological brain state. Note that classification performance is bound by the accuracy of the underlying sleep scoring as a ground truth. Since the inter-rater reliability between sleep scoring experts is typically about 80% (*Danker-Hopfe et al., 2009*), the classifier accurately predicts the experts' ratings in 80% of the time.

Second, we repeated this analysis to discriminate wakefulness from N3 sleep. Classification performance using slow wave power or spectral slope did not differ significantly (*slow wave power*: 82.09 ± 2.13%, *spectral slope*: 73.05 ± 2.97%; *permutation t-test:* p=0.054, obs. $t_{17}$ = −1.95, d = −0.72; *Figure 3b*). This shows that the spectral slope successfully discriminates wakefulness from N3 sleep despite the fact that the defining criterion of N3 sleep is pronounced slow wave activity. If LDA was used to classify all three states simultaneously (wakefulness, N3, REM; chance level = 33%) then the classifier performance was comparable for the spectral slope and slow wave power (*spectral slope*: 58.09 ± 2.35%; *slow wave power*: 63.94 ± 2.04%; *permutation t-test*: p=0.054, obs. $t_{17}$ = −1.77, d = −0.62) potentially reflecting the respective advantageous classification of either REM or N3 from wakefulness.

We repeated this analysis to discriminate anesthesia from wakefulness (n = 9). We found that classification based on the spectral slope performed better than the one based on slow wave power

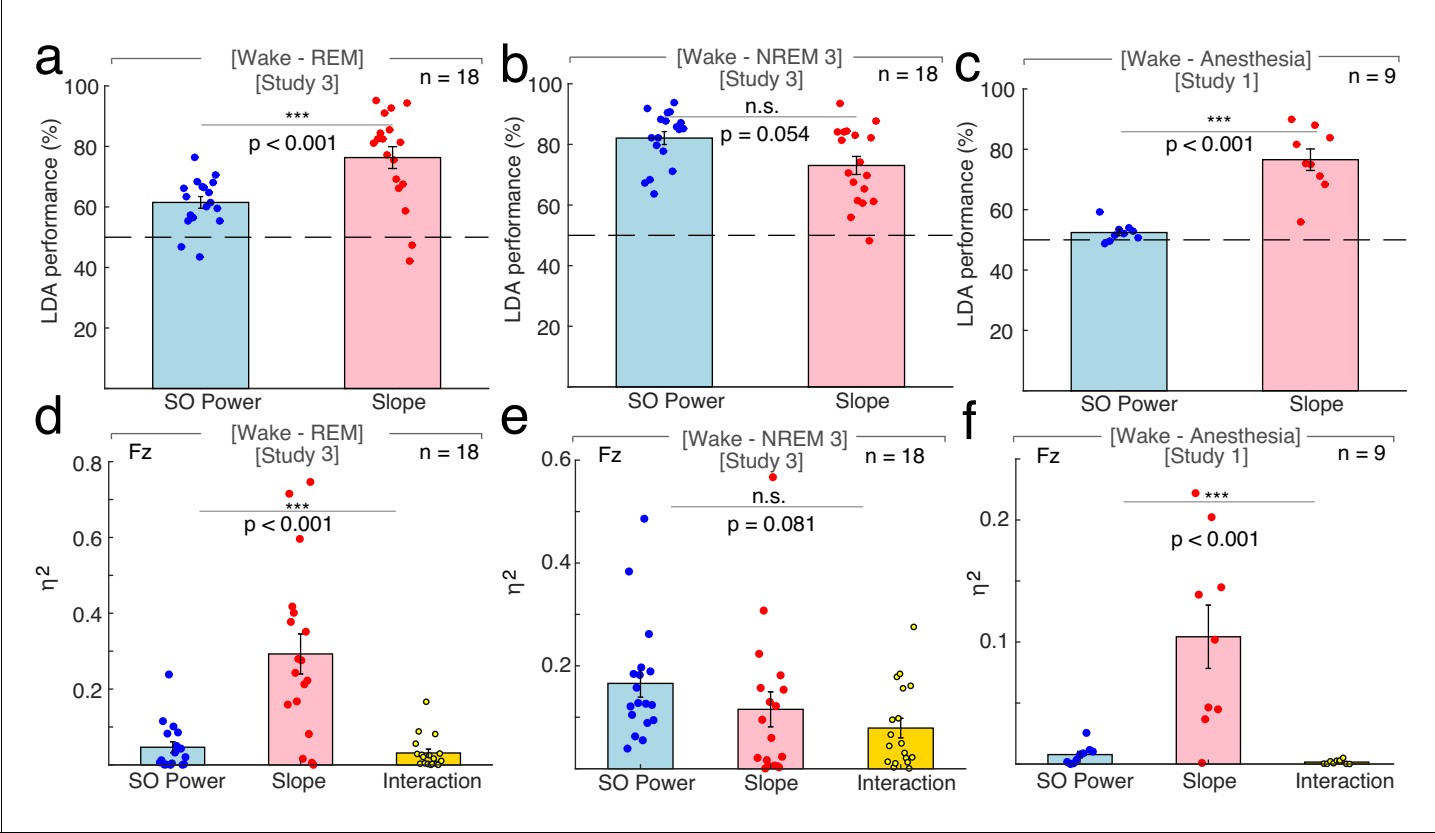

**Figure 3.** Differentiation of wakefulness from sleep and general anesthesia via Linear Discriminant Analysis (LDA) trained classifier and multivariate general linear modeling (GLM). All LDA classification performances (panel a – c) were logit-transformed and averaged across channels before comparison. (a) Using the 1/f slope (n = 18; two patients had to be excluded due to insufficient wake trials) resulted in a higher percentage of correct classification of wakefulness and REM compared to slow wave (SO) power (<1.25 Hz; *SO*: 61.50 ± 1.93% (mean ± SEM), *slope*: 76.31 ± 3.61%; *permutation t-test*: p<0.001, observed (obs.) $t_{17}$ = 3.73, d = 1.25). **p<0.01. Dashed line – chance level at 50% (*permutation t-test vs. chance*): *SO*: p<0.001, obs. $t_{17}$ = 5.51, d = 1.84, *slope*: p<0.001, obs. $t_{17}$ = 6.03, d = 2.01). (b) The use of SO power and spectral slope resulted in comparable classification of wakefulness and NREM sleep stage 3 (n = 18; *SO*: 82.09 ± 2.13%, *slope*: 73.05 ± 2.97%; *permutation t-test*: p=0.054, obs. $t_{17}$ = −1.95, d = −0.72). n.s. – not significant. Dashed line – chance level at 50% (*permutation t-tests vs. chance*): *SO*: p<0.001, obs. $t_{17}$ = 11.23, d = 3.71, *slope*: p<0.001, obs. $t_{17}$ = 6.63, d = 2.21). (c) The 1/f slope (n = 9) resulted in a higher classification accuracy of wakefulness and anesthesia with propofol compared to SO power (*SO*: 52.43 ± 1.04%, *slope*: 76.56 ± 3.56%; *permutation t-test*: p<0.001, obs. $t_8$ = 6.10, d = 2.63). ***p<0.001. Dashed line – chance level at 50% (*permutation t-test vs. chance*): *SO*: p=0.003, obs. $t_8$ = 2.33, d = 1.10, *slope*: p<0.001, obs. $t_8$ = 6.15, d = 2.89). All predictors for the multivariate GLM (panel d - f) were z-scored before modeling and calculated on data derived from scalp electrode Fz. (d) Between wakefulness and REM sleep (n = 18; two patients had to be excluded due to insufficient wake trials), the unique explained variance as quantified by eta squared ($\eta^2$) was significantly different between the 1/f slope, SO power and the interaction between the two (*slope*: 0.12 ± 0.03, *SO*: 0.17 ± 0.03, *interaction*: 0.08 ± 0.02; *repeated-measures ANOVA permutation test*: p<0.001, $F_{1.16, 19.74}$ = 19.69). ***p<0.001. (e) Variance between wakefulness and NREM sleep stage 3 (n = 18) was equally well explained by the 1/f slope, SO power and their interaction (*slope*: 0.12 ± 0.03, *SO*: 0.17 ± 0.03, *interaction*: 0.08 ± 0.02; *repeated-measures ANOVA permutation test*: p=0.081, $F_{1.72, 29.28}$ = 2.55). n.s. – not significant. f, Out of the total variation between wakefulness and general anesthesia with propofol (n = 9), significantly different proportions could be attributed to the 1/f slope, SO power and their interaction (*slope*: 0.10 ± 0.03, *SO*: 0.01 ± 0.003; *interaction*: 0.002 ± 0.001; *repeated-measures ANOVA permutation test*: p<0.001, $F_{1.01, 8.09}$ = 14.61). This effect was mainly driven by the 1/f slope. ***p<0.001.

(*spectral slope*: 76.56 ± 3.56%, *slow wave power*: 52.43 ± 1.04%; *permutation t-test*: p<0.001, obs. $t_8$ = 6.10, d = 2.63; *Figure 3c*). Note, that slow wave power was already elevated during wakefulness, which may reflect a premedication with a sedative (see *Figure 1a* and Materials and Methods).

## General Linear Model

When discerning wakefulness from REM sleep, information (as quantified by unique explained variance eta squared) about the underlying arousal state was significantly different between the spectral slope, SO power and their interaction (*repeated-measures ANOVA permutation test*: p<0.001, $F_{1.16,}$

$_{19.74}$ = 19.69). Post hoc t-tests (p-values were Bonferroni-corrected for multiple testing) revealed that this effect was predominantly driven by the spectral slope (*slope*: 0.29 ± 0.05 (mean ± SEM), *SO*: 0.05 ± 0.01; *interaction*: 0.06 ± 0.03; *post hoc permutation t-tests (Bonferroni-corrected): Slope-SO:* p<0.001, obs. $t_{17}$ = 4.29, d = 1.50, *Slope-Int.:* p<0.001, obs. $t_{17}$ = 4.84, d = 1.62; *SO-Int.:* p=0.63, obs. $t_{17}$ = 0.82, d = 0.29).

For NREM stage three sleep and wakefulness, there was no difference in explained variance between factors (*slope*: 0.12 ± 0.03, *SO*: 0.17 ± 0.03, *interaction*: 0.08 ± 0.02; *repeated-measures ANOVA permutation test*: p=0.081, $F_{1.72, 29.28}$ = 2.55; *post hoc permutation t-tests (Bonferroni-corrected): Slope-SO:* p=0.406, obs. $t_{17}$ = −1.11, d = −0.39; *Slope-Int.:* p=0.422 obs. $t_{17}$ = 1.09, d = 0.31; *SO-Int.:* p=0.032, obs. $t_{17}$ = 2.44, d = 0.88).

Between anesthesia and wakefulness, information about the state was again significantly different between factors (*slope:* 0.10 ± 0.03, *SO*: 0.01 ± 0.003; *interaction*: 0.002 ± 0.001; *repeated-measures ANOVA permutation test*: p<0.001, $F_{1.01, 8.09}$ = 14.61). As in the wake-REM differentiation, this could mainly be attributed to the spectral slope (*post hoc permutation t-tests (Bonferroni-corrected): Slope-SO:* p<0.001, obs. $t_8$ = 3.70, d = 1.75), *Slope-Int.:* p<0.001, obs. $t_8$ = 3.95, d = 1.87, *SO-Int.:* p=0.019, obs. $t_8$ = 2.67, d = 1.07).

Taken together, the results from the GLM mirrored the findings from the LDA approach: The spectral slope enabled an improved classification and contained more unique information about arousal state compared to slow wave power when differentiating wakefulness from both propofol anesthesia and REM sleep and was comparable when discerning wakefulness from N3 sleep.

## The relationship of slow waves and the spectral slope

N3 sleep and propofol anesthesia are both characterized by the occurrence of prominent slow oscillations (*Murphy et al., 2011*). Previous reports indicated that low frequency activity might serve as a marker to disentangle different arousal states (*Brown et al., 2010*; *Franks and Zecharia, 2011*; *Murphy et al., 2011*). Our results confirm and extend this observation. However, while slow wave activity (<1.25 Hz) discriminated wakefulness from N3, it was less robust in separating wakefulness from REM sleep or propofol anesthesia *Figure 3*). We conducted several control analyses to investigate the relationship of slow wave activity and the spectral slope.

First, we found that the interaction of spectral slope and slow wave activity did not explain more unique variance than the sum of the univariate metrics in a GLM, hence, indicating that the slope and slow wave activity provide complimentary information about arousal states (*Figure 3*). In addition, if lower frequencies (e.g. 1 to 20 Hz) were utilized for spectral slope estimation, MI between the hypnogram and the time-resolved spectral slope decreased (*Figure 2—figure supplement 5d*) suggesting that lower frequencies and the 30 to 45 Hz range may index distinct processes.

Second, we analyzed the changes in spectral slope during the time course of a slow wave. At the scalp EEG level, the trough of a slow wave is associated with a cortical 'down-state', while the peak reflects an 'up-state' (*Nir et al., 2011*; *Valderrama et al., 2012*). We found, that the spectral slope mirrored up-/down-states during sleep with more negative slopes observed at slow wave troughs compared to peaks (*Figure 4*). This effect was most pronounced over frontal channels (*cluster-based permutation test:* p=0.005, $d_{Trough-Peak}$ = −0.65).

Third, slow wave activity is also present to some degree during REM sleep (*Funk et al., 2016*) as well as wakefulness (*Vyazovskiy et al., 2011*). Here, we detected a significantly higher number of slow waves during N3 sleep ($SO_{N3}$ = 28.79 ± 0.79 per minute; mean ± SEM at electrode Fz) as compared to REM sleep ($SO_{REM}$ = 2.16 ± 0.89 per minute; *permutation t-test:* p<0.0001, obs. $t_{19}$ = 22.64, d = 7.05) and wakefulness ($SO_{Wake}$ = 5.05 ± 0.51 per minute; *permutation t-test*: p<0.0001, obs. $t_{19}$ = 25.32, d = 6.92; *Figure 4d*).

Interestingly, the averaged slope at the trough of the slow waves was significantly different between arousal states: −2.26 ± 0.12 in wakefulness, −3.40 ± 0.09 in N3 sleep and −4.00 ± 0.18 in REM sleep (mean ± SEM), mirroring our observation of the overall slope differences (*Figure 4c*; *permutation t-tests:* Wake-N3: p<0.0001, obs. $t_{18}$ = 7.07, d = 2.38; Wake-REM: p<0.0001, obs. $t_{18}$ = 9.67, d = 2.55; N3-REM: p=0.007, obs. $t_{19}$ = 2.73, d = 0.91).

Taken together, our control analyses indicate that slow wave activity and the spectral slope may index two distinct processes.

## Control analyses

### Evaluation of parameters for 1/f spectral slope estimation

Non-oscillatory background activity decays exponentially following a power law with a 1/f shape: $PSD(f){\sim}1/f^{\alpha}$. The spectral slope ($\alpha$) of this decay, sometimes also referred to as spectral exponent ($\beta = -\alpha$), can be estimated by a linear regression of the PSD in *log-log* space (both x- and y-axis are logarithms). In this study, we examined the spectral slope in three distinct states of reduced arousal, namely general anesthesia, NREM three and REM sleep, and in both scalp EEG and intracranial EEG recordings. We estimated the spectral slope from a linear fit to the power spectrum in *log-log* space from 30 to 45 Hz as suggested previously (*Gao et al., 2017*). There is no consensus on parameters for spectral slope estimation and a variety of settings have been employed. To address this issue, we systematically evaluated the influence of the following parameters:

1. Power
   a. Method for power calculation
   Using a Multitaper approach (*Prerau et al., 2017*) for power estimation resulted in a better signal to noise ratio in sleep compared to a single Hanning taper, a periodogram or Welch's method (no overlap, single taper; *Figure 2—figure supplement 6*) across all examined frequencies (0.5 to 45 Hz; Fig X, $p < 0.001$).
   b. Segment length
   A change in segments length from 10 to 30 s under anesthesia or 30 to 10 s in sleep did not change the overall observed pattern of spectral slopes (*Figure 1—figure supplement 3b,c*; *Figure 2—figure supplement 7b,c*) and estimates from both segment lengths were strongly correlated ($p<0.0001$; *Figure 1—figure supplement 3d*; *Figure 2—figure supplement 7d*).
   c. Reference scheme
   Bilateral linked mastoids, common average, Laplacian and clinical bipolar reference schemes resulted in comparable spectral slope patterns with more negative slopes for sleep than for rest (*Figure 2—figure supplement 8b*). Although absolute slopes values varied slightly, they were strongly correlated between montages ($p<0.001$; *Figure 2—figure supplement 8c*).
2. Frequency range
   a. Center frequency for fit
   We evaluated the relationship between hypnogram and time-resolved slope as a function of different center frequencies (±10 Hz around center frequency, starting from 20 up to 150 Hz) and found that spectral slope estimates only correlated significantly/had significant Mutual Information (MI) with the hypnogram if center frequencies up to 40 Hz were selected for the fit (*Figure 2—figure supplement 5a*).
   b. Length of fit
   We evaluated the relationship between hypnogram and time-resolved slope as a function of fit lengths (from 30 Hz onwards with a 10 Hz increase of fit length up to 100 Hz). The results showed that spectral slopes estimates could be fitted with variable fit length from 30 Hz onwards and still resulted in a significant correlation/MI with the hypnogram (*Figure 2—figure supplement 5b*).
   c. Fit to low frequencies
   We explored fits to lower frequencies in both propofol anesthesia and sleep. Under anesthesia, spectral slope estimates from fits to 1 to 40 and 30 to 45 Hz resulted in a similar pattern with more negative slopes during anesthesia compared to wakefulness (*Figure 1—figure supplement 4c*). Effect sizes between states and goodness of fits were comparable in both frequency ranges (*Figure 1—figure supplement 4e*) while classification performance between states was better for the lower frequency fit (*Figure 1—figure supplement 4f*), possibly due to including frequency bands that exhibit strong differences between wakefulness and anesthesia (e.g. delta/alpha oscillation; *Purdon et al., 2013*).
   For sleep, we evaluated fits to lower frequencies starting from 1 to 5 Hz with an increasing length of additional 5 Hz per fit after discounting the oscillatory components from the power spectrum by means of irregular resampling (IRASA; *Wen and Liu, 2016a*, *Figure 2—figure supplement 5c*). When comparing the MI between the spectral slope fits to a random distribution derived from a block swapping procedure, the 30 to 45 frequency range resulted in significantly higher MI than the fits to lower frequencies (*Figure 2—figure supplement 5d*).
3. Fit

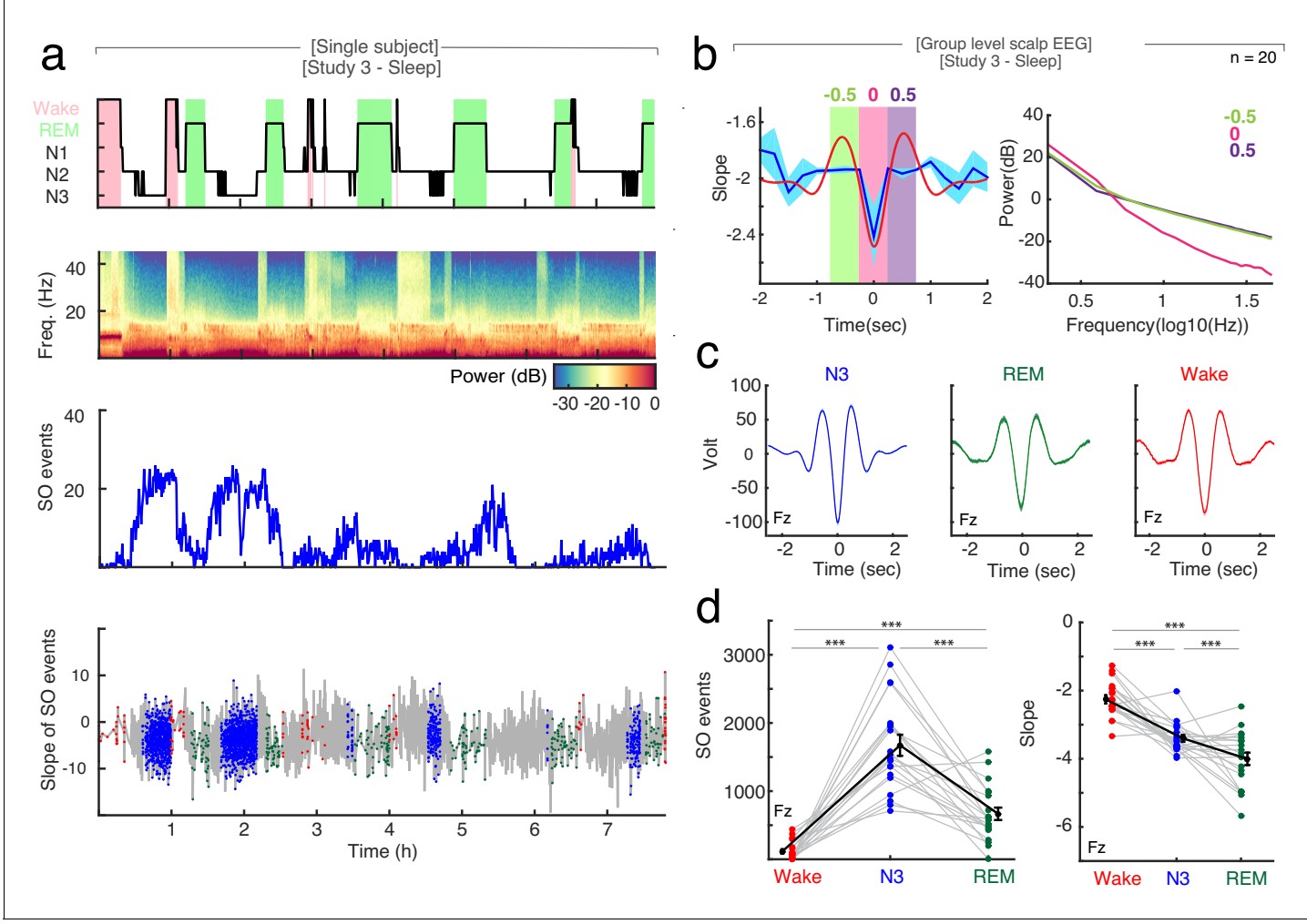

**Figure 4.** The relationship between spectral slope and slow waves in sleep. (a) Single subject example: Upper panel: Hypnogram. Wake periods are highlighted in pink, REM periods in light green. Upper middle panel: Multitapered spectrogram of electrode Fz. Lower middle panel: Number of slow wave (SO) events during 30 s segments of sleep in electrode Fz. Note the decreasing number of SO events during the course of the night. Lower panel: Spectral slope of SO events occurring in N3 (blue), wakefulness (red) and REM sleep (green) in electrode Fz. Background: Time-resolved slope of electrode Fz in light gray. (b) Right panel: Average spectral slope changes over the time course of all slow waves in scalp EEG (n = 20) during sleep (blue; mean ± SEM); superimposed in red is the average slow wave of all subjects. Highlighted are the following 0.5 s time windows relative to the slow wave trough: −750 to −250 (center −0.5 s; green), −250 to 250 (center 0 s; pink) and 250 to 750 ms (center 0.5 s; purple). Left panel: Power spectra in log-log space within specified time windows during the slow wave: −750 to −250 (center: −0.5 s; green), −250 to 250 (center: 0 s; pink) and 250 to 750 ms (center: 0.5 s; purple). Note the steep power decrease during the trough of the slow wave (pink). (c) Group level (n = 20) average waveforms in electrode Fz during N3 (blue), REM sleep (green) and wakefulness (red; mean ± SEM). (d) Left: Slow wave events per minute in wakefulness (red), N3 (blue) and REM (green) in scalp EEG channel FZ (n = 20). In black mean ± SEM. *Permutation t-tests*: ***p<0.001. Right: Slope of slow wave events on the group level (n = 20; averaged across all 19 EEG electrodes) in wakefulness (red), N3 (blue) and REM sleep (green). Mean ± SEM in black. *Permutation t-tests*: ***p<0.001.

a. Linear regression

We compared a linear regression with the MATLAB *polyfit.m* function to the *eBOSC* algorithm (*Caplan et al., 2001*; *Kosciessa et al., 2020a*; *Whitten et al., 2011*) which employs MATLAB's *robustfit.m* function. While both algorithms resulted in slightly different absolute slopes estimates (anesthesia: *Figure 1—figure supplement 4a,b*; sleep: *Figure 2— figure supplement 9a*), both estimates revealed the same overall pattern with a more negative slope for sleep compared to rest and anestehsia compared to wakefulness. Moreover, slope estimates derived from both algorithms were strongly correlated (p<0.0001; *Figure 1—figure supplement 4d*; *Figure 2—figure supplement 9c*) and did

not differ in effect size (*Figure 1—figure supplement 4e*) or their goodness of fit to the power spectrum (*Figure 2—figure supplement 9d*).

 b. Model fit

 We further compared a linear regression with polyfit (see above) and the model fit of the *FOOOF* algorithm (*Haller et al., 2018*). Both algorithms resulted in similar slope estimates (anesthesia: *Figure 1—figure supplement 4a,b*; sleep: *Figure 2—figure supplement 9a*) and followed the overall slope pattern with a more negative slope for sleep compared to rest and anesthesia compared to wakefulness. Moreover, slope estimates derived from both algorithms were strongly correlated (p<0.0001; *Figure 1—figure supplement 4d*, *Figure 2—figure supplement 9b*) and did not differ in effect size (*Figure 1—figure supplement 4e*) or their goodness of fit to the power spectrum (*Figure 2—figure supplement 9d*).

Taken together, the pattern of a more negative slope during sleep and anesthesia compared to wakefulness was robustly observed across a wide spectrum of parameters. A Multitaper approach (*Prerau et al., 2017*) to calculate the power spectral density was characterized by a higher signal-to-noise ratio in comparison to other methods (*Figure 2—figure supplement 6*). The choice of segment length depends on the cortical state where quasi-stationarity can be assumed (*Figure 1—figure supplement 3*, *Figure 2—figure supplement 7*). Here, we observed slope effects on very different timescales ranging from milliseconds (*Figure 4*) to full night recordings (*Figure 2*). The reference scheme did not have a significant effect on the overall observed pattern (*Figure 2—figure supplement 8*) and can be selected depending on the precise research question: e.g. a bipolar or Laplacian reference might be more suited to examine local phenomena. In sleep, center frequencies from 20 Hz up to 40 Hz (±10 Hz) and fit length of 20 Hz or more (from 30 Hz onwards) exhibited a significant relationship with the hypnogram (*Figure 2—figure supplement 5a,b*). Spectral slope estimated from fits to lower frequencies e.g. 1 to 20 Hz, on the other hand, had a significantly lower MI with the hypnogram than the 30 to 45 Hz frequency range (*Figure 2—figure supplement 5d*). Under anesthesia, both fits to 1 to 40 and 30 to 45 Hz led to a comparable slope pattern with more negative slopes under anesthesia compared to wakefulness (*Figure 1—figure supplement 4*). Thus, while the 30 to 45 Hz frequency range is well suited to differentiate wakefulness from both sleep and anesthesia, other frequency ranges might be advantageous when examining only one state (e.g. lower frequency fits under anesthesia than in sleep). The use of different slope fitting algorithms (polyfit, robustfit (*eBOSC*), *FOOOF*) did not impact the overall observed slope pattern in both sleep and under anesthesia and the derived slope values were strongly correlated (*Figure 1—figure supplement 4*, *Figure 2—figure supplement 9*). Hence, all three algorithms can be used interchangeably in the examined states and frequency ranges. A model fit via e.g. the *FOOOF* algorithm (*Haller et al., 2018*) might be the preferred choice when a bend in the PSD (also called 'knee') is observed.

## The relationship of connectivity and the spectral slope

Rodent studies suggest that fronto-parietal theta and high-gamma network connectivity correlates with arousal levels in both sleep and general anesthesia (*Pal et al., 2018*; *Pal et al., 2016*). We tested this notion and directly compared connectivity estimates to the spectral slope metric: We found that the spectral slope was superior to fronto-parietal theta connectivity in tracking sleep stages and in reliably differentiating REM and N3 sleep (*Figure 2—figure supplement 10*). Note that our dataset did not have a sufficient number of intracranial electrodes in the parietal lobe to analyze fronto-parietal connectivity since the parietal lobe is an infrequent site for clinical exploration for epilepsy. Hence, we restricted our analyses to theta-band connectivity in scalp EEG. Future studies will be needed to address the relationship of high gamma-band connectivity and the spectral slope.

## Discussion

Our results demonstrate that the spectral slope, which reflects one parameter describing the aperiodic component of the electrophysiological power spectrum, facilitates the reliable discrimination of wakefulness from propofol anesthesia, NREM and REM sleep. Here, we present results from four

independent studies providing converging evidence that the spectral slope constitutes a marker that tracks arousal levels in humans.

## Neurophysiological markers of arousal states

Consciousness is commonly assessed on two axes – *content* (e.g. the experience) and *level* (e.g. vigilance; *Boly et al., 2013*; *Laureys, 2005*). While conscious *content* is thought to fluctuate during sleep, mostly in the form of dreams during REM (*Siclari et al., 2017*), the *arousal level* is generally reduced as compared to wakefulness. Both components are typically judged by verbal report of the research subject or patient leading to the approximation that *content* is equivalent to conscious experience as related by the subject whereas the *arousal level* corresponds to the subject's ability to respond. Notably, there are some obvious restrictions to these definitions (e.g. the reduced arousal level in REM sleep prevents the subject from relating his experience to the experimenter unless awakened), which make more objective electrophysiological measures desirable.

Several neurophysiological metrics of conscious *content* such as the Perturbational Complexity Index (PCI; *Casali et al., 2013*) have been introduced. While PCI is decreased in N3 sleep and GABAergic (e.g. propofol) anesthesia, it resembles wakefulness during REM sleep and ketamine anesthesia, which are both associated with vivid dreams (*Casali et al., 2013*; *Pal et al., 2015*; *Siclari et al., 2017*). A recent EEG study under anesthesia with propofol, xenon and ketamine found that the PCI correlated with the spectral exponent derived from the 1 to 40 Hz frequency range (*Colombo et al., 2019*). A related study reported that the spectral slope derived from the 0.5 to 35 Hz frequency range became progressively steeper from wakefulness to REM sleep, N2 and N3 sleep (*Miskovic et al., 2019*). Critically, these metrics did not reliably differentiate *arousal levels*, i.e. they did not generalize to distinguishing wakefulness from REM sleep.

The overall slowing of EEG activity and the occurrence of oscillations in lower frequency bands has previously been linked to reduced *arousal levels* (e.g. slow waves and spindles in sleep [*Prerau et al., 2017*], delta waves and alpha oscillation under propofol anesthesia [*Purdon et al., 2013*]). REM sleep, also called 'paradoxical' sleep (*Siegel, 2011*), is characterized by a 'wake-like' EEG without prominent oscillations in humans. Differentiating between wakefulness and REM solely from the electrophysiological brain state has been challenging and to date still requires simultaneous EMG and EOG recordings to detect muscle atonia and rapid eye movements (*Iber et al., 2007*).

Here, we demonstrate that the non-oscillatory, aperiodic part of the power spectrum, which is devoid of prominent low-frequency oscillatory components and can be approximated by the 1/f decay of the power spectrum estimated from the 30 to 45 Hz frequency range, reliably differentiates wakefulness from all three states of reduced *arousal level*, namely REM, N3 sleep and general anesthesia with propofol.

## The neurophysiologic basis of 1/f dynamics

1/f dynamics are observed across a variety of tasks (*He et al., 2010*; *Miller et al., 2009a*; *Miller et al., 2009b*; *Voytek et al., 2015*), change with lifespan (*Voytek et al., 2015*), and exhibit state-dependent variations during sleep (*Freeman and Zhai, 2009*; *Leemburg et al., 2018*; *Miskovic et al., 2019*; *Robinson et al., 2011*) and anesthesia (*Colombo et al., 2019*; *Gao et al., 2017*). Critically, these dynamics can be observed irrespective of the employed recording modality and species (*Colombo et al., 2019*; *Freeman and Zhai, 2009*; *Gao et al., 2017*; *He et al., 2010*; *Leemburg et al., 2018*; *Miskovic et al., 2019*). However, to date, the underlying neural mechanisms giving rise to the prominent 1/f decay of the electrophysiological power spectrum are not well understood (*Buzsáki et al., 2012*; *He et al., 2010*; *Miller et al., 2009a*; *Pesaran et al., 2018*).

For instance, it had been observed that broadband activity (~2 to 150 Hz) and high frequency power (>80 Hz) correlate with population neuronal firing rates in macaques and humans (*Ray and Maunsell, 2011*; *Manning et al., 2009*) as well as task performance across a range of behavioral experiments (*Honey et al., 2012*; *Miller et al., 2009b*; *Miller et al., 2014*). Furthermore, several lines of research indicate that the spectral slope not only tracks the overall firing rate (*Buzsáki et al., 2012*; *Miller et al., 2009a*) but also correlates with a variety of related phenomena, including metrics that can be derived from electrophysiological time-series analysis such as entropy (*Miskovic et al., 2019*), ensemble synchronization (*Shen et al., 2003*) or signal complexity (*Pereda et al., 1998*).

Moreover, a link to the local balance between excitation and inhibition (E/I-balance; *Gao et al., 2017*) as well as dendritic filtering (*Buzsáki et al., 2012*) has been suggested previously.

Notably, the EEG power spectrum in *log-log* space does not follow a straight line with a constant spectral exponent, but is characterized by the occurrence of so called 'knees' (bends). Different knee frequencies have been reported, such as around 1 ~ 2 Hz (*He et al., 2010*),~20 Hz (*Robinson et al., 2011*; *Robinson et al., 2001*) and ~75 Hz (*Miller et al., 2009a*). The characteristic form of the power spectrum can be modeled by the multiplication of Lorentzian functions (*He, 2014*; *He et al., 2010*; *Miller et al., 2009a*; *Miller et al., 2014*) where the knee frequencies are directly related to the time constants of the exponential decay (e.g. 2 ~ 3 ms for the 75 Hz knee and ~100 ms for the 1 ~ 2 Hz knee). Although the origin of these time constants remains unclear, it has been suggested that the 2 ~ 3 ms time constant originates from synaptic currents, while the 100 ms time constant reflects membrane leak (*He, 2014*; *Miller et al., 2009a*). Other studies attributed the 20 Hz knee to the low pass filtering properties of dendrites (*Robinson et al., 2011*; *Robinson et al., 2001*). Critically, the 1/f dynamics in between those knee frequencies exhibit different spectral exponents: While an exponent of 2–3 is observed for center frequencies (~1 to 80 Hz; *Freeman and Zhai, 2009*; *He et al., 2010*; *Miller et al., 2009a*), higher frequencies (80 to 500 Hz) exhibited an exponent closer to 4 (*Miller et al., 2009a*), thus, suggesting that the spectral slopes in different frequency ranges might be the consequence of different generative mechanisms.

A recent study proposed that the shape of the power spectrum of human intracranial EEG is the product of local (fast) and distributed (slow) recurrent networks that are balancing excitation and inhibition in such a way that the network is tuned to the edge of dynamic instability, thus, maximizing information processing capacities (*Chaudhuri et al., 2018*). Further in-silico modeling of E/I-balance suggested that the spectral slope in the 30 to 50 Hz range functions as an index of this balance where an increase in inhibition is accompanied by a decrease of spectral slope (*Gao et al., 2017*). One testable hypothesis that arises from these observations is that cell-type-specific causal manipulations by e.g. optogenetics through selective targeting of excitatory pyramidal or inhibitory parvalbumin- or somatostatin-positive interneurons should bias the spectral slope in opposite directions.

Future studies involving single neuron recordings and optogenetic manipulation will be needed to unravel the precise relationship between population firing statistics and changes in the spectral slope. Comparative studies in rodents (*Gao et al., 2017*; *Leemburg et al., 2018*), non-human primates (*Gao et al., 2017*) and humans (*Colombo et al., 2019*; *He et al., 2010*; *Miller et al., 2009a*; *Miller et al., 2009b*; *Miskovic et al., 2019*) combined with modeling work (*Chaudhuri et al., 2018*; *Robinson et al., 2011*; *Robinson et al., 2001*) has the potential to integrate the divergent findings into a coherent framework, which is critical to further elucidate the neurophysiologic basis of 1/f dynamics and their relationship to arousal levels.

## Functional significance of 1/f dynamics in arousal states

Here, we found a decreased 1/f slope in N3 sleep, REM sleep and under general anesthesia compared to wakefulness in four independent studies. If the spectral slope reflects the local E/I-balance (*Gao et al., 2017*), then our results would indicate that decreased arousal states are characterized by increased inhibition. In support of this consideration, several previous studies reported that propofol anesthesia and N3 sleep are associated with increased inhibition (*Brown et al., 2011*; *Gao et al., 2017*; *Timofeev et al., 2001*) as well as prominent changes in neuronal firing rates (*Lewis et al., 2012*; *Timofeev et al., 2001*; *Vyazovskiy et al., 2009*; *Watson et al., 2016*). Notably, there is little evidence supporting a similar association for REM sleep. However, a recent two-photon calcium imaging study in rodents reported a reduction of the overall cortical firing rate during slow-wave and even further during REM sleep (*Niethard et al., 2016*). Critically, this study also demonstrated that reduced firing rates during REM sleep were accompanied by a selective increase of inhibitory parvalbumin interneuron activity, thus, reflecting a relative shift towards inhibition (*Niethard et al., 2016*). These results parallel the pattern as observed in this present study (*Figure 2—figure supplement 11*) where the steepest decay of the power spectrum (the most negative PSD slope) occurred during REM sleep.

Jointly, these findings imply that REM sleep could be associated with the highest level of cortical inhibition, thus, resulting in a steeper decay of the power spectrum. This notion of increased inhibition during REM sleep offers a likely mechanistic explanation for certain REM-defining phenomena, such as muscle atonia (*Scammell et al., 2017*) or the clinical observation that epileptic seizures

during the night predominantly occur out of more excitable, highly synchronized NREM sleep and only rarely out of less excitable, desynchronized REM sleep (*Ng and Pavlova, 2013*).

## Practical considerations for analyzing 1/f dynamics

A wide array of settings has been employed to examine 1/f features in electrophysiological recordings. Practical guidelines on how to select parameters for spectral slope estimation remain scare. Here, we explored a range of different settings for a series of parameters to reliably estimate 1/f activity in arousal states.

The estimation of 1/f background activity first requires a spectral decomposition of the underlying time-series. Here, we directly compared several algorithms (multiple slepian tapers, single Hanning window, Periodogram, Welch's method) and found that the Multitaper approach yielded the highest signal-to-noise ratio (*Figure 2—figure supplement 6*). Critically, window length and spectral smoothing parameters need to be adjusted to a reasonable time window where stationarity of the time series can be assumed (e.g. 30 s segments for sleep; *Figure 2—figure supplement 7*). The choice of different reference schemes or slope fitting algorithms, namely a linear regression (see Methods), the eBOSC algorithm (*Kosciessa et al., 2020a*) as well as the FOOOF algorithm (*Haller et al., 2018*), resulted in a comparable overall spectral slope pattern with strongly correlated slope estimates (*Figure 1—figure supplement 4*, *Figure 2—figure supplement 9*).

Additionally, we also explored a range of different fit frequencies (*Figure 1—figure supplement 4*, *Figure 2—figure supplement 5*). Collectively, our results indicate that the spectral slope should be approximated in a frequency range where no pronounced oscillatory activity and no prominent bend ('knee') are present. Therefore, the optimal choice of frequency range might differ between arousal states.

In sleep, fitting the power spectrum to frequencies above 30 Hz (and <45 Hz to avoid the European line noise frequency at 50 Hz) shared the highest Mutual Information with the hypnogram (*Figure 2—figure supplement 5*). An inclusion of lower frequencies (e.g. 10 to 30 Hz or various fits length from 1 Hz onwards (compare *Figure 2—figure supplement 5a,d*) diminished this relationship, indicating a conflation of background and oscillatory activity. Critically, our results from a general linear model demonstrated that the slope in higher frequencies (above 30 Hz) carried independent information from slow oscillatory activity about the current brain state (*Figure 3*).

Incorporating frequencies below 20 Hz resulted in a better separation between wakefulness and anesthesia (*Figure 1—figure supplement 4*), possibly due to the inclusion of the alpha range (~8–10 Hz) where prominent oscillations are observed under propofol anesthesia (*Figure 1*).

Taken together, a fit to the 30 to 45 frequency range reliably differentiated wakefulness from anesthesia, NREM three and REM sleep.

## Conclusions

Collectively, our results from four independent studies provide five main advances: First, the spectral slope tracks changes in arousal levels in both propofol anesthesia and sleep in humans with high temporal precision and is observable on a wide range of timescales from sub-second epochs to full night recordings.

Second, our results provide empirical evidence for the notion that the slope in the range from 30 to 45 Hz correlates well with all stages of arousal. Previous studies analyzing the spectral slope during states of reduced arousal have either employed different frequency ranges (*Colombo et al., 2019*; *Freeman and Zhai, 2009*; *He et al., 2010*; *Miskovic et al., 2019*) or focused either on anesthesia (*Colombo et al., 2019*; *Gao et al., 2017*) or sleep (*Miskovic et al., 2019*; *Pereda et al., 1998*; *Shen et al., 2003*). Here, we demonstrate that this frequency range is well suited to track arousal in both anesthesia and different sleep levels including REM and NREM sleep.

Third, the spectral slope mirrored the rapid changes in excitability observed over the course of a slow wave directly tracking cortical 'up-' and 'down-states', hence, providing a mechanistic link between the spectral slope and population synchrony during slow oscillations.

Fourth, our observations support the premise that anesthesia is a brain-wide state (*Brown et al., 2010*), whereas sleep exhibits network-specific activity patterns. Here, we observed that sleep dependent slope modulations were strongest in the medial temporal lobe and medial PFC, two key

regions for sleep-dependent memory consolidation (*Diekelmann and Born, 2010*; *Preston and Eichenbaum, 2013*; *Stickgold and Walker, 2013*).

Fifth, the spectral slope can be reliably estimated from scalp EEG recordings, thus, providing an accessible marker that can easily be incorporated into intraoperative neuromonitoring or automatic sleep stage classification algorithms. In the future, this marker could potentially be utilized to monitor other states of reduced arousal such as epileptic seizures, coma, the vegetative or minimally conscious state.

# Materials and methods

**Key resources table**

| Reagent type (species) or resource | Designation | Source or reference | Identifiers | Additional information |
|---|---|---|---|---|
| Software, algorithm | MATLAB Release 2015a and 2018a; Signal Processing Toolbox, Curve Fitting Toolbox, Statistics and Machine Learning Toolbox | The MathWorks Inc, 2020, Natick, Massachusetts, USA | ID_source: SCR_001622 | |
| Software, algorithm | Adobe Illustrator CS6 and CC 2018 | Adobe Inc, 2020, Dublin, Republic of Ireland | ID_source: SCR_010279 | |
| Software, algorithm | FieldTrip 20170829 | *Oostenveld et al., 2011*; *Stolk et al., 2018* | ID_source: SCR_004849 | http://www.fieldtriptoolbox.org/ |
| Software, algorithm | EEGLAB_14_0_0b | *Delorme and Makeig, 2004* | ID_source: SCR_007292 | https://sccn.ucsd.edu/eeglab/index.php |
| Software, algorithm | FreeSurfer 5.3.0 | *Dale et al., 1999*; *Fischl, 2012* | ID_source: SCR_001847 | https://surfer.nmr.mgh.harvard.edu/ |
| Software, algorithm | LCMV Beamformer | *Van Veen et al., 1997* | | |
| Software, algorithm | IRASA (Irregular Resampling Auto-Spectral Analysis) | *Wen and Liu, 2016b* | | |
| Software, algorithm | Multitaper Spectral Analysis | *Prerau et al., 2017* | | |
| Software, algorithm | FOOOF (Fitting Oscillations and One Over F) | *Haller et al., 2018* | | https://pypi.org/project/fooof/ |
| Software, algorithm | eBOSC (extended Better OSCillation detection) | *Kosciessa et al., 2020b* | | https://github.com/jkosciessa/eBOSC |
| Software, algorithm | Mutual Information | *Quian Quiroga and Panzeri, 2009* | | http://prerau.bwh.harvard.edu/multitaper/ |
| Software, algorithm | BioSig Toolbox - Logit transformation | *Schlogl and Brunner, 2008* | ID_source: SCR_008428 | |
| Software, algorithm | General Linear Model | *Siegel et al., 2015* | | |
| Software, algorithm | Slow wave detection | *Helfrich et al., 2018*; *Staresina et al., 2015* | | |
| Software, algorithm | Random Block Swapping (statistics) | *Canolty et al., 2006*; *Aru et al., 2015* | | |
| Software, algorithm | Connectivity | iPLV (imaginary Phase Lock Value) - *Nolte et al., 2004*; rhoortho (orhtogonalized power correlation) - *Hipp et al., 2012* | | |
| Other | | NA | | |

## Participants

We collected four independent datasets for this study to assess the neurophysiological basis of states of reduced arousal, namely general anesthesia and sleep. We recorded either non-invasive scalp electroencephalography (EEG) or intracranial EEG (electrocorticography; ECoG) using subdural grid and strip electrodes and stereotactically placed depth electrodes (SEEG; for coverage see *Figure 1—figure supplement 1*).

## Anesthesia

The EEG and intracranial anesthesia studies were conducted at the University Hospital of Oslo. All participants or their parents provided informed written consent according to the local ethics committee guidelines (Regional Committees for Medical and Health Research Ethics in Oslo case number 2012/2015 and extension 2012/2015–8) and the Declaration of Helsinki.

### Study 1 - Anesthesia scalp EEG

Ten patients (two female) undergoing anterior cervical discectomy and fusion participated in Study one and received a total intravenous anesthesia with remifentanil and propofol. They had an American Society of Anesthesia status of I - III, were between 46 and 64 years old (53.3 ± 5.7 years; mean ± SD) and otherwise healthy. Data were recorded from the induction of anesthesia to the recovery from 25 channel EEG according to the 10–20 layout (EEG Amplifier, Pleasanton, California, USA) with an additional row of electrodes (F9, F10, T9, T10, P9, P10) at a digitization rate of 512 Hz, or in the case of one patient at 256 Hz. The electrode for referencing was placed at CP1. Three patients were not recorded for the planned entire time span – one recording was only started after induction, while two were stopped before recovery (*Juel et al., 2018*).

### Study 2 - Anesthesia intracranial EEG

A total of 12 patients (three female) with intractable epilepsy participated in Study 2. They were between 8 and 52 years old (26.6 ± 13.2 years; mean ± SD). Data were collected during the explantation of the intracranial electrodes from induction of anesthesia up to the point of their removal. All patients received total intravenous anesthesia with propofol and remifentanil at the University Hospital of Oslo. All patients were placed back on their usual antiepileptic medication before the procedure. Data were recorded on a Natus NicoletOne system with a 128-channel capacity and a digitization rate of 1024 Hz for up to 64 or 512 Hz for up to 128 channels.

## Anesthetic management

All patients received a premedication with 3.75 to 7.5 mg midazolam (Dormicum, Basel, Switzerland); the anesthesia scalp EEG group (Study 1) received additional 1 g oral paracetamol (Paracet, Weifa, Oslo, Norway) as well as 10 mg oxycodone sustained release tablet (OxyContin, Dublin, Ireland) for postoperative pain management. Propofol (Propolipid, Fresenius Kabi, Uppsala, Sweden) and remifentanil (Ultiva, GlaxoSmithKline, Parma, Italy) were administered by computer-controlled infusion pumps (B Braun Perfusor Space, Melsungen, Germany) using a target-controlled infusion (TCI) program (Schnider for propofol and Minto for remifentanil) in order to achieve plasma concentrations sufficient for anesthesia and analgesia. Prior to start of anesthesia all patients received an infusion of Ringer's-Acetate (5 ml /kg) to prevent hypotension during anesthesia induction, as well as 3–5 ml 1% lidocaine intravenously to prevent pain during propofol injection. All patients were preoxygenated with 100% oxygen and received the non-depolarizing muscle relaxant cisatracurium for intubation (Nimbex, GlaxoSmithKline, Oslo, Norway). After intubation the inspiratory oxygen fraction was reduced to 40%; nitric oxide was not used.

## Sleep

### Study 3 - Sleep scalp EEG

Study three was conducted at the University of California at Berkeley. All participants were informed and provided written consent in accordance with the local ethics committee (Berkeley Committee for Protection of Human Subjects Protocol Number 2010-01-595). We analyzed recordings from 20 young healthy participants (20.4 ± 2.0 years, mean ± SD; 12 females). Polysomnography was recorded during an 8 hour period as well as during 5 min quiescent rest with eyes closed before and

after sleep. Data were recorded on a Grass Technologies Comet XL system (Astro-Med, Inc, West Warwick, RI) with a 19-channel EEG using the standard 10–20 setup as well as three electromyography (EMG) and four electro-oculography (EOG) electrodes at the outer canthi. The EEG was referenced to the bilateral linked mastoids and digitized at 400 Hz (0.1 to 100 Hz; *Helfrich et al., 2018*; *Mander et al., 2015*; *Mander et al., 2014*; *Mander et al., 2013*). Sleep staging was carried out by trained personnel (B.A.M.) and according to recent guidelines (*Iber et al., 2007*).

### Study 4 - Sleep intracranial EEG
Study four was conducted at the University of California at Irvine, Medical Center. Ten epilepsy patients (six females) undergoing invasive pre-surgical localization of their seizure focus were included in this study. All patients provided informed consent according to the local ethics committees of the University of California at Berkeley and at Irvine (University of California at Berkeley Committee for the Protection of Human Subjects Protocol Number 2010-01-520; University of California at Irvine Institutional Review Board Protocol Number 2014–1522, UCB relies on UCI Reliance Number 1817) and gave their written consent before data collection. They were between 22 and 55 years old (33.1 ± 11.5 years; mean ± SD). Electrode placement was solely dictated by clinical criteria (Ad-Tech, SEEG: 5 mm inter-electrode spacing; Integra, Grids: 1 cm, 5 or 4 mm spacing). Data were recorded with a Nihon Kohden recording system (256 channel amplifier, model JE120A), analogue-filtered above 0.01 Hz and digitally sampled at 5 kHz. To facilitate gold-standard sleep staging, simultaneous EOG, electrocardiography (ECG) from five leads and EEG was recorded by exemplary electrodes of the 10–20 setup depending on the localization of the intracranial electrodes but mostly consisting of Fz, Cz, C3, C4 and Oz. A surrogate EMG signal was derived from the ECG and EEG by high-pass filtering above 40 Hz. Sleep staging was carried out by trained personnel (B.A.M.) according to recent guidelines (*Iber et al., 2007*).

## Data preprocessing
### Study 1 - Anesthesia scalp EEG
Data were imported into FieldTrip (*Oostenveld et al., 2011*) and epoched in 10 s segments. An Independent Component Analysis (*fastica; Hyvärinen, 1999*) was used to clean the data from systematic artifacts such as the ECG. Further data cleaning was done manually after inspection by a neurologist (R.T.K.) and an anesthesiologist (J.D.L). On average, the patients had 1183 ± 81.42 ten-second epochs of which 196 ± 103.19 were marked as noisy (15.81 ± 3.15%). No channels were excluded or interpolated. Data were referenced using the common average, demeaned and detrended. Wake periods were defined as time before induction and after anesthesia when the patients responded reliably to verbal commands of the study personnel. Anesthesia periods were defined as time after induction until the termination of propofol application.

### Study 2 – Anesthesia intracranial EEG
Data were recorded with a 512 Hz digitization rate in eight patients. Four additional patients were recorded with a digitization rate of 1024 Hz and these datasets were down-sampled to 512 Hz. Data were then imported to FieldTrip (*Oostenveld et al., 2011*), epoched into ten-second segments and inspected by a neurologist (R.T.K.) for epileptic activity and then manually cleaned of epileptic and other non-neural artifacts. The awake state was defined as time before start of propofol, anesthesia was defined as time after loss of consciousness (unresponsiveness to verbal commands assessed by study personnel and attending anesthetist). After fusing the pre-implantation T1-weighted MRI and the post-implantation Computer Tomography (CT) scans, electrodes were automatically localized by an openly available brain atlas (Freesurfer; *Fischl, 2012*) using the FieldTrip toolbox (*Stolk et al., 2018*) and cross-validated by independent manual inspection by two neurologists (R.T.K.; R.F.H.). Contacts in white matter or lesions were discarded. The remaining signals were then bipolar referenced to their lateral neighbor, demeaned and detrended.

### Study 3 - Sleep scalp EEG
The EEG was referenced to bilateral linked mastoids and data were imported to EEGLAB (*Delorme and Makeig, 2004*) and epoched into 5 s segments. Epochs that contained artifacts (e.g. eye blinks or movement) were manually inspected and rejected by a trained scorer (B.A.M.). None of

the channels were discarded or interpolated. On average, the participants had 5748.9 ± 10.01 of these five second epochs and 946.95 ± 542.68 of them were rejected (16.44 ± 2.98%), comparable to the anesthesia scalp EEG recordings. The data from the healthy sleep participants have been reported before and were cleaned in a comparable approach (*Helfrich et al., 2018*; *Mander et al., 2015*; *Mander et al., 2014*; *Mander et al., 2013*). For further analysis in MATLAB (MATLAB Release R2018b, The MathWorks, Inc, Natick, Massachusetts, United States), the data were then imported into FieldTrip (*Oostenveld et al., 2011*).

### Study 4 – Sleep intracranial EEG

Data were imported to FieldTrip (*Oostenveld et al., 2011*), downsampled to 500 Hz and segmented into 30 s segments for subsequent data analysis. Anatomical localization was carried out by fusing pre-implantation T1-weighted Magnetic Resonance Imaging (MRI) scans with post-implantation MRI and both automatic and manual labeling of the electrode position (*Stolk et al., 2018*). As above, epileptic, white matter and channels with other artifacts were discarded. The data were bipolar referenced, demeaned and detrended.

## Spectral analysis

(1)To obtain average power spectra, after artifact removal the data were epoched into ten-second segments for anesthesia and 30 s segments for sleep. (2) Time-frequency decomposition was accomplished using a Fast Fourier Transformation (*mtmfft*, FieldTrip (*Oostenveld et al., 2011*) from 0.5 Hz to 45 Hz in 0.5 Hz steps. The analysis was limited to 45 Hz due to line noise at 50 Hz in the Oslo recordings and then adopted to all consecutive studies for consistency. To obtain reliable spectral estimates we utilized a Multitaper approach (*Prerau et al., 2017*) based on discrete prolate slepian sequences (*dpss*; anesthesia: 9 tapers for 10 s segments, no overlap, frequency smoothing of ± 0.5 Hz; sleep: 29 tapers for 30 s segments, no overlap, frequency smoothing of ± 0.5 Hz). (3) The power spectrum of each state was averaged over all samples of the state (wake and anesthesia or rest, wake, non-rapid eye movement sleep stage 3 (N3) and rapid eye movement sleep (REM)), channels and subjects (*Figure 1b,c* and *Figure 2b,c*). For better comparison, we visualized the effect at the scalp level. For Study 2 simultaneous EEG recordings were not available.

For the control analysis of the influence of segment length and number of tapers on the spectral slope estimate, anesthesia EEG recordings were epoched into 30 s segments, while sleep was epoched into 10 s segments. Time frequency decomposition was again done with FieldTrip's *mtmfft* with a frequency smoothing of ± 0.5 Hz and *dpss* tapers resulting in 29 for anesthesia and nine for sleep. Note that the number of tapers is a direct result of the choice of segment length and frequency smoothing (*Prerau et al., 2017*):

$$\text{Number of } dpss \text{ tapers} = (2*(\frac{Segment\ length\ (sec)*frequency\ smoothing\ (Hz)}{2}))-1$$

## Comparison of Multitaper, Single-taper, Periodogram and Welch's Method

For the comparison of the signal-to-noise ratios of different power calculations (*Figure 2—figure supplement 6*), the Multitaper spectral decomposition (*Prerau et al., 2017*) was calculated as outlined above. For all further computations, sleep data were epoched into 30 s segments.

For the Single-taper analysis, power was calculated using the Fast Fourier Transformation *mtmfft* of FieldTrip (*Oostenveld et al., 2011*) with no overlap after applying a Hanning taper. Spectral estimates were obtained between 0.5 and 45 Hz in 0.5 Hz steps.

For the calculation of the Periodogram, we used MATLAB's *periodogram.m* function (MATLAB and Signal Processing Toolbox Release R2018b, MathWorks, Inc, USA) and the default rectangular window and number of points as defined by the sampling rate of 400 Hz. To get the same number of observations as in the Multi- and Single-taper approaches, the frequencies closest to the ones chosen in the Multi- and Single-taper approach above (0.5 to 45 Hz in 0.5 Hz steps) were chosen for further analysis.

For the Welch's method, power was calculated with a 30 s window size and no overlap resulting in a single Hamming window employing MATLAB's *pwelch.m* function (MATLAB and Signal Processing Toolbox Release R2018b, MathWorks, Inc, USA). The number of points were defined by the

sampling rate. Again, the frequencies closest to the Multi-/Single-taper approach were selected for further analysis to enable a direct comparison of signal-to-noise ratios (SNR).

SNR was calculated by dividing the average power of each frequency at every channel by the standard deviation of this frequency and channel.

## Spectral slope estimation

We calculated the spectral slope by fitting a linear regression line to the power spectrum in *log-log* space between 30 and 45 Hz, since it had been shown previously that this range correlates best with changes in arousal in rodents and monkeys, as well as with different excitation-inhibition ratios in simulations (*Gao et al., 2017*). In line with previous reports, we excluded the low frequencies that contain strong oscillatory activity, which may distort the linear fit as well as the range over 50 Hz, which is confounded by both line noise (50 Hz in Europe, 60 Hz in the US) as well as broadband muscle artifacts.

We then adapted this range to the calculation of the slope in the other studies for consistency reasons. To compute a time resolved estimate of the spectral slope, we calculated the best line fit to the 10 (anesthesia) or 30 (sleep) second segments of the Multitapered power spectra (see above) in *log-log* space using polynomial curve fitting (*polyfit.m,* MATLAB and Curve Fitting Toolbox Release R2015a, The MathWorks, Inc, Natick, Massachusetts, United States). One subject in Study 3 (sleep EEG) exhibited an excessive noise level during wakefulness; therefore, his data had to be excluded from all slope comparisons to wakefulness.

## Control analyses of spectral slope estimation

For the evaluation of different center frequencies for the linear regression, the spectral slope was estimated from intracranial sleep recordings (Study 4) as a function of different center frequencies (±10 Hz around center frequency starting from 20 up to 150 Hz) and different fit lengths (from 30 Hz onward with a 10 Hz increase of fit length up to 100 Hz) using the same procedure as outlined above (*Figure 2—figure supplement 5*).

Moreover, the spectral slope was estimated from low frequencies in different recordings: (1) Anesthesia scalp EEG (Study 1) in the 1 to40 Hz range as previously reported (*Colombo et al., 2019*) and (2) in Sleep scalp EEG (Study 3) from 1 to 5 Hz with an increasing length of additional 5 Hz per fit after discounting the oscillatory components from the power spectrum by means of irregular resampling (IRASA; *Wen and Liu, 2016a*; *Figure 2—figure supplement 5c*). The rationale for using this method was to exclude strong low frequency oscillations, such as slow waves, which could possibly distort the slope estimate.

Furthermore, we estimated the spectral slope from the 30 to45 Hz range using two additional algorithms, the FOOOF package (*Haller et al., 2018*) and the eBOSC algorithm (*Kosciessa et al., 2020a*) in both anesthesia and sleep scalp EEG (Study 1 and 3). While eBOSC performs a linear regression in *log-log* space using MATLAB's *robustfit.m*, FOOOF estimates both the oscillatory and aperiodical part of the power spectrum with a model fit: First an exponential fit is performed to the power spectrum in *semi-log* space, then this fit is subtracted from the power spectral density (PSD). The residual signal is treated as a combination of oscillations and noise and is repeatedly fit with multiple Gaussian fits to detect possible oscillatory peaks until the noise floor is reached. The detected peaks are then validated against the mixed oscillatory/noise signal and subtracted from the original PSD. The new residual signal is then fit again for a better estimate of the aperiodic signal. Both, the Multi-Gaussian fits and the improved aperiodic fit are then combined into a model (compare to *Figure 3* (*Haller et al., 2018*). When the PSD does not contain a bend, also called knee, in the aperiodic signal, then the algorithm is equivalent to fitting a line in *log-log* space (*Haller et al., 2018*).

## Mutual Information

Mutual Information (MI) is an information theoretical metric, which quantifies the mutual dependence of the two signals, specifically the amount of information gained about one variable when observing the other (*Quian Quiroga and Panzeri, 2009*). This is particularly useful for non-linear, binned signals. Mutual information between the two signals X and Y was defined as

$$\mathrm{MI(X;Y)} = \sum_{x \in X}\sum_{y \in Y} \mathrm{p(x,y)} * \log_2\left(\frac{\mathrm{p(x,y)}}{\mathrm{p(x)} * \mathrm{p(y)}}\right)$$

where p(x,y) depicts the joint probability function and p(x) and p(y) indicate the class probabilities. Probabilities were normalized by their sum. For MI analysis (*Figure 2b*, *Figure 2—figure supplements 2*, *5*, *8* and *10*), we epoched the time-resolved slope into 30 s segments (the hypnogram was staged in 30 s epochs) and discretized it into five bins (Wake, REM, N1, N2, N3) using the *discretize. m* function of MATLAB Signal Processing Toolbox Release R2015a (MathWorks Inc, USA). Mutual Information was calculated using the *MutualInformation.m* function from MATLAB Central File Exchange (*Dwinell, 2010*).

## Beamformer analysis

We source-localized the slope difference between wakefulness and sleep in scalp EEG (Study 3, n = 19; one patient had to be excluded due to insufficient wake trials; *Figure 2—figure supplement 3*). Cortical sources of the sensor-level EEG data were reconstructed by using a LCMV (linearly constraint minimum variance) beamforming approach (*Van Veen et al., 1997*) to estimate the time series for every voxel on the grid. A standard T1 MRI template and a BEM (boundary element method) headmodel from the FieldTrip toolbox (*Oostenveld et al., 2011*) were used to construct a 3D template grid at 1 cm spacing in standard MNI space. Electrode location was derived from a standard 10/20 template from the FieldTrip toolbox (*Oostenveld et al., 2011*). Prior to source projection, sensor level data were common average referenced and epoched into 30 s segments. To minimize computational load, we selected a two second data segment from the center of every epoch to construct the covariance matrix. The LCMV spatial filter was then calculated using the covariance matrix of the sensor-level EEG data with 5% regularization. Spatial filters were constructed for each of the grid positions separately to maximally suppress activity from all other sources. The resulting time courses in source space then underwent spectral analysis using a Fourier transform after applying a Hanning window. The PSD slope of every segment was then estimated using linear regression in the range from 30 to 45 Hz as outlined before. Mean slope values during sleep were subtracted from the mean slope during wakefulness at every voxel in source space and then source-interpolated onto a standard template brain in MNI space.

We employed a regions-of-interest (ROI) based approach focusing on prefrontal cortex (PFC) subregions since a reliable source localization of medial temporal lobe (MTL) activity using 19-channel EEG remains challenging. In line with our results from the intracranial sleep study (Study 4, *Figure 2C* and *Figure 2—figure supplement 4*), we defined the mPFC and dorsolateral PFC (dlPFC) as ROI. To identify ROI at source level, we used the Automated Anatomical Labeling (AAL) atlas as implemented in FieldTrip (*Oostenveld et al., 2011*). The ROI mPFC encompassed the following regions: 23 and 24 – 'Frontal_Sup_Medial_L and R', 25 and 26 – 'Frontal_Med_Orb_L and R', 27 and 28 – 'Rectus_L and R', 31 and 32 – Cingulum_Ant_L and R'. The ROI dlPFC consisted of the following atlas tissue labels: 3 and 4 – 'Frontal_Sup_L and R', 7 and 8 – 'Frontal_Mid_L and R', 11 and 12 – 'Frontal_Inf_Oper_L and R', 13 and 14 – 'Frontal_Inf_Tri_L and R' and 15 and 16 'Frontal_Inf_Orb_L and R'.

## Classification analysis

We employed a linear discriminant analysis (LDA) to assess if slow wave power or the spectral slope were a better predictor of wakefulness or sleep (*Figure 1—figure supplement 4*, *Figure 3a–c*). We utilized a leave-one-exemplar-out cross-validation approach that was repeated 50 times after randomly sampling an equal number of sleep and REM trials to equate the number of samples (*classify. m*, MATLAB and Statistics and Machine Learning Toolbox Release R2015a, MathWorks Inc, USA). Then every sample of the subsampled distribution was held out of the training dataset once. The LDA classifier was trained on the remaining samples and tested on the held-out test sample. The classifier performance was then assessed as percent correct. Two of the 20 sleep EEG participants had to be excluded due to insufficient number of wake trials. LDA performances were logit-transformed (*logit.m* from the BioSig toolbox (*Schlogl and Brunner, 2008*) and averaged across channels prior to statistical comparison.

## Multivariate General Linear Model

In order to model the unique contributions as well as the interaction between slow wave power (<1.25 Hz) and spectral slope in the prediction of arousal states, we first scaled both parameters by means of a z-score and then calculated a general linear model using the MATLAB *fitlm.m* function (MATLAB and Statistics and Machine Learning Toolbox Release R2018b, MathWorks Inc, USA). This method offered the possibility to model both the main effects as well as the interaction between the two predictors (*Figure 3d–f*). Critically, we utilized an unbalanced design, which implicitly orthogonalizes the contribution of different factors and hence, distills non-overlapping unique explained variance (*Siegel et al., 2015*), which was subsequently quantified by means of the effect size (eta squared).

## Spectral slope estimation during a slow wave

Slow wave events (*Figure 4*, *Figure 2—figure supplement 10*) were detected for each channel based on established algorithms (*Helfrich et al., 2018*; *Staresina et al., 2015*): The raw signal was bandpass-filtered between 0.16 and 1.25 Hz and zero crossings were detected. Events were then selected using a time (0.8 to 2 s duration) and an amplitude criterion (75% percentile). The raw data were then epoched relative to the trough of the slow wave (±2.5 s). Time-frequency decomposition was computed in 500 ms time windows with a 250 ms overlap using FieldTrip (*Oostenveld et al., 2011*) (*mtmfft*, frequency smoothing of ±2 Hz and one dpss taper). The spectral slope was calculated by the best line fit in these time windows in *log-log* space between 30 and 45 Hz using polynomial curve fitting (*polyfit.m*, MATLAB and Curve Fitting Toolbox Release R2015a, MathWorks Inc, USA).

## Statistical testing

The spectral slope of the awake and anesthetized state was compared using Student's t-test for paired samples (*Figure 1b,c*). The observed t-values were then compared against a surrogate distribution after re-calculating the test statistic 10,000 times with randomly shuffled labels (called *permutation t-test* in the manuscript). Post hoc t-tests were Bonferroni-corrected for multiple comparisons.

To compare three states (awake, NREM and REM), we utilized Greenhouse-Geisser corrected 1-way repeated measures analysis of variance (*Figure 2b and c*; RM-ANOVA). We adopted an approach similar to the *permutation t-test*, where the original RM-ANOVA output was compared to a shuffled distribution after randomly flipping condition labels 10,000 times (*repeated-measures ANOVA permutation test*). Effect size was calculated using Cohen's d.

To assess the spatial extent of the observed effects in EEG, we calculated cluster-based permutation tests to correct for multiple comparisons as implemented in FieldTrip (*Oostenveld et al., 2011*) (Monte-Carlo method; maxsize criterion; 1000 iterations). A permutation distribution was obtained by randomly shuffling condition labels and then compared to the actual distribution to obtain an estimate of significance. Spatial clusters are formed by thresholding independent t-tests of slope differences between wake and anesthesia (*Figure 1b*) or wake and sleep (*Figure 2b*) at a p value < 0.05. All results were Bonferroni-corrected for multiple comparisons. In order to control for EMG as a potential confound in the sleep EEG (Study 3), we utilized a partial correlation (Spearman) that partialled the slope of the EMG out of the correlation before computing the cluster-based permutation test (*Figure 2—figure supplement 2b*). Correlation coefficients (r-values) were transformed into t-values using the following formula (N = number of subjects):

$$t = \frac{r * sqrt(N - 2)}{sqrt(1 - r^2)}$$

For statistical assessment of the Mutual Information (MI), we employed surrogate testing (*Figure 2b*, *Figure 2—figure supplement 2a*). To obtain a surrogate distribution from the observed data, we utilized a random block swapping procedure (*Aru et al., 2015*; *Canolty et al., 2006*). The number of repetitions was equal to the number of available sleep stages. On every iteration, we re-calculated the MI of these block swapped hypnograms with the discretized time-resolved slope to create a surrogate distribution against which we could compare our original observation. To compare the results across subjects, we z-scored the values by subtracting the mean of the surrogate distribution from the observed MI and dividing by the standard deviation of the surrogate distribution. Note that a z = 1.96 reflects an uncorrected two-tailed p-value of 0.05, while a z-score of >2.8

indicates a Bonferroni-corrected significant p-value (p<0.05/19 channels=0.0026). The z-values were transformed into p-values for topographic depiction (*Figure 2b*; *Figure 2—figure supplement 2a*) based on a normal cumulative distribution function (two-tailed).

## Connectivity

For the analysis of fronto-parietal connectivity (*Figure 2—figure supplement 10*), we choose electrode Fz and Pz in our sleep EEG recordings (Study 3; n = 20) to calculate the magnitude squared coherence from frequencies of 0.1 to 64 Hz in 0.1 Hz steps using the *mscohere.m* function from the MATLAB Signal Processing Toolbox (Release R2015a, MathWorks, Inc, USA) as described previously (*Pal et al., 2016*; *Pal et al., 2015*). Note that coherence estimates reflect both power changes as well as changes in phase synchrony. Therefore, we also calculated the Phase-Locking Value (PLV) and amplitude correlations (rho) to disentangle the effects of phase and power, respectively. To discount the effects of volume spread, we calculated the imaginary PLV (*Nolte et al., 2004*) (iPLV) and orthogonalized power correlations (*Hipp et al., 2012*) (rhoortho).

We then quantified the Mutual Information (MI; see above) to compare how well the results capture the changes between different sleep stages across the night (*Quian Quiroga and Panzeri, 2009*). For this analysis we only utilized the slope values of electrode Fz (as we were calculating the other measures in Fz-Pz) and defined theta from 4 to 10 Hz analog to *Pal et al., 2016*; *Pal et al., 2015*.

## Acknowledgements

This work was supported by Grant LE 3863/2-1 (JDL) and HE 8329/2–1 (RFH) of the German Research Foundation (Deutsche Forschungsgemeinschaft), a National Institute of Neurological Disorders and Stroke Grant R37NS21135 (RTK), the Hertie Foundation (Network of Excellence in Clinical Neurosciences; RFH), R01AG03116408 (MPW), RF1AG05401901 (MPW), RF1AG05410601 (MPW) and F32-AG039170 (BAM), all from the National Institute of Health. We thank Jie Zheng, Julia Kam and the EEG technicians at UC Irvine Medical Center for their assistance and all the patients for their participation.

## Additional information

### Funding

| Funder | Grant reference number | Author |
| --- | --- | --- |
| Deutsche Forschungsge-meinschaft | LE 3863/2-1 | Janna Desiree Lendner |
| National Institute of Neurolo-gical Disorders and Stroke | R37NS21135 | Robert T Knight |
| Deutsche Forschungsge-meinschaft | HE 8329/2-1 | Randolph F Helfrich |
| National Institute of Mental Health | R01AG03116408 | Matthew P Walker |
| National Institute of Mental Health | RF1AG05401901 | Matthew P Walker |
| National Institute of Mental Health | RF1AG05410601 | Matthew P Walker |
| National Institute of Mental Health | F32-AG039170 | Bryce A Mander |

The funders had no role in study design, data collection and interpretation, or the decision to submit the work for publication.

### Author contributions

Janna D Lendner, Conceptualization, Data curation, Software, Formal analysis, Funding acquisition, Validation, Investigation, Visualization, Methodology, Writing - original draft, Project administration,

Writing - review and editing; Randolph F Helfrich, Software, Project administration, Writing - review and editing; Bryce A Mander, Resources, Data curation, Methodology; Luis Romundstad, Resources, Project administration; Jack J Lin, Resources, Investigation, Project administration; Matthew P Walker, Resources, Data curation, Investigation, Project administration, Writing - review and editing; Pal G Larsson, Resources, Data curation, Investigation, Project administration; Robert T Knight, Conceptualization, Resources, Supervision, Funding acquisition, Methodology, Project administration, Writing - review and editing

### Author ORCIDs
Janna D Lendner ORCID https://orcid.org/0000-0002-1967-6110
Randolph F Helfrich ORCID https://orcid.org/0000-0001-8045-3111

### Ethics
Human subjects: We collected four independent datasets for this study to assess the neurophysiological basis of states of reduced arousal, namely sleep and general anesthesia. Study 1 - Anesthesia scalp EEG: All participants were informed and provided written consent in accordance with the local ethics committee (Regional Committees for Medical and Health Research Ethics in Oslo case number 2012/2015 and extension 2012/2015-8). Study 2 - Anesthesia intracranial EEG: All participants were informed and provided written consent in accordance with the local ethics committee (Regional Committees for Medical and Health Research Ethics in Oslo case number 2012/2015 and extension 2012/2015-8). Study 3 - Sleep scalp EEG: All participants were informed and provided written consent in accordance with the local ethics committee (Berkeley Committee for Protection of Human Subjects Protocol Number 2010-01-595). Study 4 - Sleep intracranial EEG: All patients provided informed consent according to the local ethics committees of the University of California at Berkeley and at Irvine (University of California at Berkeley Committee for the Protection of Human Subjects Protocol Number 2010-01-520; University of California at Irvine Institutional Review Board Protocol Number 2014-1522, UCB relies on UCI Reliance Number 1817) and gave their written consent before data collection.

### Decision letter and Author response
Decision letter https://doi.org/10.7554/eLife.55092.sa1
Author response https://doi.org/10.7554/eLife.55092.sa2

## Additional files
### Supplementary files
• Transparent reporting form

### Data availability
Lendner, Janna et al. (2020), An electrophysiological marker of arousal level in humans, v3, UC Berkeley, Dataset.

The following dataset was generated:

| Author(s) | Year | Dataset title | Dataset URL | Database and Identifier |
|---|---|---|---|---|
| Lendner JD | 2020 | An electrophysiological marker of arousal level in humans, v3, UC Berkeley, Dataset | https://doi.org/10.6078/D1NX1V | Dryad Digital Repository, 10.6078/D1NX1V |

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
