## [Decision Letter]

**Acceptance summary:**

This study relies on both invasive and non-invasive EEG recordings in humans who were either awake, anesthetized or sleeping, and shows that a particular marker of neural activity – the slope of the EEG power spectrum – can be used to distinguish between different states of arousal. Importantly, this measure can be reliably estimated from scalp EEG recordings, and accurately separates REM sleep from wakefulness, which to date had been challenging using EEG. Furthermore, the authors show that anesthesia reflects a brain-wide state, while sleep patterns are observed in specific networks.

**Decision letter after peer review:**

Thank you for submitting your article "An Electrophysiological Marker of Arousal Level in Humans" for consideration by *eLife*. Your article has been reviewed by three peer reviewers, one of whom is a member of our Board of Reviewing Editors, and the evaluation has been overseen by Laura Colgin as the Senior Editor. The following individual involved in review of your submission has agreed to reveal their identity: Giovanni Piantoni (Reviewer #3).

The reviewers have discussed the reviews with one another and the Reviewing Editor has drafted this decision to help you prepare a revised submission. In recognition of the fact that revisions may take longer than the two months we typically allow, until the research enterprise restarts in full, we will give authors as much time as they need to submit revised manuscripts.

Summary:

Lendner and colleagues assess 1/f EEG activity as a proxy of arousal by comparing both invasive and non-invasive recordings of awake, anesthetized and sleeping humans. The authors calculated the slope in the log-log power spectral density during periods of wakefulness, N3 and REM sleep and propofol anesthesia. They consistently found across four studies (both intracranial and scalp EEG) that the slope is shallower for wakefulness as compared to the other states of reduced arousal. A series of control analyses (LDA, changes in center frequency and width of the frequency range, link to slow waves) confirmed the main finding. The findings are exciting and well tested over multiple studies and recording setups. Particularly intriguing is the similarity between propofol and sleep.

Although the study in general represents an important contribution to the field of human electrophysiological research, we are not entirely convinced by the used methods, inferential statistics, and by the way the findings are integrated into the existing literature. The analysis of multiple data sets in order to draw inferences on various levels is compelling and impressive but warrants a more consistent methodological approach and the acknowledgement of earlier findings.

Essential revisions:

1) Overall, the manuscript is lacking connection with existing literature, leading to unwarranted novelty claims and rather unspecific hypotheses. Authors should integrate the existing literature more thoroughly.

This is especially of importance for the Introduction and Discussion sections. For example, the authors cite Colombo et al., 2019, in the Introduction but do not acknowledge that this study performed a comparison of EEG spectral exponents between the awake state and anesthesia under different drugs, including Propofol, and found a steepening of the spectrum under Propofol anesthesia compared to awake rest. Similarly, the authors cite Miskovic et al., 2019 without noting that the cited study compared PSD exponents between different sleep stages and found results comparable to those of the current manuscript.

It is essential to put the results of the current study into context and note that they replicate earlier findings and are not the first of their kind in data from human subjects. The appeal of novelty claims in the Introduction which states that only non-human animal results exist for the central question of the manuscript, should be omitted. Instead, we suggest to formulate more precise hypotheses regarding the direction of effects (steepening vs. flattening of spectra) and to touch upon E/I balance and neurotransmitters already in the Introduction.

2) We had several concerns regarding the inferential statistics used. Permutation-based or non-parametric tests are more appropriate for the majority of performed tests and should be implemented.

More specifically: Inferential statistics should be adapted to meet the distributional properties of the data they are performed on. The comparison of very small samples (e.g. n = 9) using t-tests should be replaced by non-parametric or permutation-based approaches. Similarly, comparing goodness of fit statistics using t-tests should be omitted (subsection “The relationship of slow waves and the spectral slope”) and replaced by proper model comparisons.

Additionally, performance correct percentages of the used LDA should undergo an appropriate transformation before being tested against each other (e.g. logit).

On a more general note, we suggest to replace the LDA approach with a multivariate model predicting arousal states based on the predictors the authors aim to compare. This way their predictive power can be compared directly (if scaled in the same way, e.g. z-scored) and their shared variance in the dependent variable can be accounted for. Furthermore, the discussion of prediction accuracies should include a reference to the "ground truth" that is predicted here, the ratings of a trained specialist which are 80% reliable.

3) One of the major claims of the paper is that the effect of interest is consistent across techniques (EEG, iEEG) and brain states (propofol, sleep). To support this claim, the authors would need to be consistent in their analysis pipelines of the different datasets. More specifically:

– Why were different length segments (and taper numbers) used for sleep vs anesthesia? Would be better for direct comparison to keep these parameters the same.

– The iEEG studies and the scalp EEG studies use a very different reference. Bipolar for iEEG and common average for the scalp EEG studies (the reference is not clearly specified for study 3 though, please clarify). One would expect that the choice of reference would have a large impact on the slope, but this does not seem the case. For example, local dynamics captured by bipolar referencing are in general very different from global effects such as those observed in scalp EEG. In fact, the Laplacian reference does not change the results. Could the authors discuss in more detail the role of referencing in their interpretation of the neurophysiological basis of 1/f dynamics?

4) Furthermore, we had some concerns regarding anatomical claims.

The fact that sleep stage-slope correlation seems to be dominated by medial PFC/temporal sites is interesting, though might be biased by the iEEG montage, as those are also the regions with the most electrodes due to epilepsy monitoring. There should be a statistical analysis that shows that those regions have a significantly higher proportion of "differentiating" electrodes than other brain regions. In addition to a sound statistical analysis, the authors should provide a table with the number of total electrodes and the number of electrodes showing a significant change in slope, per brain region. As it is, it's hard to define the regions of interest based on the brain plots shown.

Additionally, it would have been much more powerful if the anatomical claims could be corroborated by the EEG recordings. Not sure source localization is possible on the EEG data given low electrode numbers, but that would really have strengthened the story. In current form it remains a little unclear what the main sources of all these effects are.

5) The fitting of 1/f spectral exponents should be improved and frequency ranges should be motivated more clearly.

Although it is correct that Gao et al., 2017, used a frequency range of 30-50 Hz for their 1/f PSD fit, this choice was motivated by the linear decrease of power in LFP recordings within that frequency range. However, oscillatory parts of the spectrum, whether of neural or non-neural origin (line noise), can aptly be fitted and excluded from a 1/f fit using the FOOOF package (https://fooof-tools.github.io/fooof/). The use of this or another software package (e.g. BOSC or eBOSC: https://github.com/jkosciessa/eBOSC) would allow the authors to directly asses the change of PSD exponents across different levels of arousal across wider frequency ranges without having to alter time-series first using IRASA. In fact, Colombo and colleagues show a steepening of the PSD between 1 and 40 Hz from awake rest compared to Propofol anesthesia, a result the authors could try to replicate and discuss using the proposed analyses techniques. Furthermore, goodness of fit statistics for the performed linear fits are missing and should be added for any kind of fitting procedure the authors decide to apply.

6) Discussion could be more in depth in terms of:

– connection with existing literature;

– underlying mechanisms/generation of the recorded signals;

– interpretation of the findings and implications for our understanding of sleep vs wake processes;

– actual discussion of practical use, i.e., how reliable is slope as a measure to distinguish these states on the single subject level.

[Editors' note: further revisions were suggested prior to acceptance, as described below.]

Thank you for resubmitting your work entitled "An Electrophysiological Marker of Arousal Level in Humans" for further consideration by *eLife*. Your revised article has been evaluated by Laura Colgin (Senior Editor) and Saskia Haegens (Reviewing Editor).

The manuscript has been improved but there are some remaining issues that need to be addressed before acceptance, as outlined below:

We appreciate the very thorough revisions, especially methods wise, and believe they have strengthened the paper considerably. After discussion with one of the original reviewers, we have one remaining point we would still like to see clarified, but other than that all our previous concerns have been adequately dealt with.

The issue we are still unclear on pertains to this claim:

"Critically, the accuracy of this classification is comparable to trained personnel, since the inter-rater reliability between sleep state scoring experts is typically about 80%".

However, if the ground truth here consists of the sleep staging done by trained experts, for whom we know the inter-rater reliability is typically about 80%, then a 100% accuracy of the classifier would be a 100% match with these expert raters (i.e., not necessarily a 100% accurate classification of sleep stage). In other words, the performance of the classifier is bound by the accuracy of the ground truth. Since we do not know the "true" sleep stage, we can only conclude that the classifier accurately predicts the experts' ratings 80% of the time here. Please add some language to make this explicit in the discussion of these results.

---

## [Author Response]

Essential revisions:1) Overall, the manuscript is lacking connection with existing literature, leading to unwarranted novelty claims and rather unspecific hypotheses. Authors should integrate the existing literature more thoroughly.This is especially of importance for the Introduction and Discussion sections. For example, the authors cite Colombo et al., 2019, in the Introduction but do not acknowledge that this study performed a comparison of EEG spectral exponents between the awake state and anesthesia under different drugs, including Propofol, and found a steepening of the spectrum under Propofol anesthesia compared to awake rest. Similarly, the authors cite Miskovic et al., 2019, without noting that the cited study compared PSD exponents between different sleep stages and found results comparable to those of the current manuscript.It is essential to put the results of the current study into context and note that they replicate earlier findings and are not the first of their kind in data from human subjects. The appeal of novelty claims in the Introduction which states that only non-human animal results exist for the central question of the manuscript, should be omitted. Instead, we suggest to formulate more precise hypotheses regarding the direction of effects (steepening vs. flattening of spectra) and to touch upon E/I balance and neurotransmitters already in the Introduction.

We apologize about the lack of connection to existing literature. We thank the reviewers for their comments and carefully revisited both the Introduction and Discussion. The revised manuscript features a completely rewritten Introduction and Discussion section with a clearer structure. Critically, we now introduce the concept of E/I-balance early in the Introduction and clearly state our predictions with regard to the hypothesized link between E/I-balance and 1/f features.

Introduction:

“General anesthesia is a reversible, pharmaceutically induced state of unconsciousness, while sleep is internally generated and cycles between rapid (REM) and non-rapid eye movement sleep (NREM; (Brown et al., 2010; Franks and Zecharia, 2011a). […]While previous studies that included lower frequency power in their slope estimates, found the slope of REM to be close to wakefulness (He et al., 2010), we specifically investigated if the aperiodic background activity in the 30 – 50 Hz range could reliably discriminate REM sleep from wakefulness and NREM sleep.”

2) We had several concerns regarding the inferential statistics used. Permutation-based or non-parametric tests are more appropriate for the majority of performed tests and should be implemented.More specifically: Inferential statistics should be adapted to meet the distributional properties of the data they are performed on. The comparison of very small samples (e.g. n = 9) using t-tests should be replaced by non-parametric or permutation-based approaches.We thank the reviewers for this suggestion and now employ non-parametric permutation statistics throughout the manuscript. In the previous version of the manuscript, we utilized cluster-based permutation tests when assessing the spatial extent of the data (see e.g. Figure 1B, 2B, Figure 2—figure supplement 2).In the revised manuscript, we replaced t-tests by permutation t-test in which the original observation was compared against a surrogate distribution after re-calculating the test statistic (e.g. T- or F-values) 10,000 times with randomly shuffled condition labels. The empirically observed t-value was then compared relative to the surrogate distribution of permuted t-values (e.g. Figure 1B, C; Figure 3A-C). For the repeated measures ANOVA, we adopted a similar approach where the original RM-ANOVA output was compared to a shuffled distribution after randomly flipping condition labels 10,000 times (e.g. Figure 2B, C; Figure 3D-F). Post-hoc t-tests were Bonferroni corrected for multiple comparisons.Similarly, comparing goodness of fit statistics using t-tests should be omitted (subsection “The relationship of slow waves and the spectral slope”) and replaced by proper model comparisons.

We agree with the reviewers and removed the goodness of fit statistic comparison using t-test from the manuscript. Instead we now evaluate the shared variance using a multivariate general linear model:

“First, we found that the interaction of spectral slope and slow wave activity did not explain more unique variance than the sum of the univariate metrics in a GLM, hence, indicating that the slope and slow wave activity provide complimentary information about arousal states (Figure 3). In addition, if lower frequencies (e.g. 1 – 20 Hz) were utilized for spectral slope estimation, MI between the hypnogram and the time-resolved spectral slope decreased (Figure 2—figure supplement 5D) suggesting that lower frequencies and the 30 – 45 Hz range may index distinct processes.”

Additionally, performance correct percentages of the used LDA should undergo an appropriate transformation before being tested against each other (e.g. logit).

We thank the reviewers for this suggestion and performed logit-transformation of the LDA correct percentages before comparison. We edited Figure 3 and revised the results in the manuscript accordingly.

Results:

**“**The spectral slope discriminates wakefulness from states of reduced arousal.

Our findings provide evidence that the spectral slope reliably discriminates wakefulness from sleep. Multiple prior reports indicated that slow waves are a hallmark of decreased arousal states (Brown et al., 2010; Franks and Zecharia, 2011a; Murphy et al., 2011). […] Note, that slow wave power was already elevated during wakefulness, which may reflect a premedication with a sedative (see Figure 1A and Materials and methods).”

On a more general note, we suggest to replace the LDA approach with a multivariate model predicting arousal states based on the predictors the authors aim to compare. This way their predictive power can be compared directly (if scaled in the same way, e.g. z-scored) and their shared variance in the dependent variable can be accounted for.

We thank the reviewers for this recommendation. Accordingly, we calculated a multivariate model to predict the arousal states based on the slope and slow oscillation (SO; < 1.25 Hz) power. This approach enabled us to directly compare the unique variance that was explained by either the spectral slope or SO power. Moreover, this method facilitated a direct assessment of the multivariate interaction of both parameters.

In order to model the unique contributions of the parameters as well as their interaction, we first scaled both by means of a z-score and then calculated their main effects as well as their interaction using a multivariate general linear model. Critically, we utilized an unbalanced design, which implicitly orthogonalizes the contribution of different factors and hence, distills non-overlapping unique explained variance, which was subsequently quantified by means of the eta squared metric.

**Author response image 1. sa2fig1:** Uni- and multivariate discrimination of wakefulness and REM sleep using the spectral slope and slow oscillation power. Single subject example at scalp EEG electrode Fz. a, While both 1/f slope (z-scored; left panel) and slow oscillation power (SO, < 1.25 Hz; z-scored; right panel) are able to differentiate wakefulness (red) from REM sleep (green; black: mean ± SEM) in a univariate comparison, the 1/f slope provides most of the discriminative power in a multivariate space as seen in panel b (SO power vs. 1/f slope; both z-scored). Note that the states (red – wakefulness; green – REM sleep) are clearly discernible based on the first feature plotted on the x-axis (the spectral slope) while taking SO power (on the y-axis) into account, does not contribute independent information.

In order to better illustrate our approach, we added Author response image 1, which depicts the relationship of uni- and multivariate differences between wakefulness and REM sleep in one subject: While both the z-scored spectral slope and the z-scored SO power discriminate wake and REM in a univariate comparison (panel a), the multivariate display of the same data (panel b) does not contribute additional information. Note that the main discriminating factor, which successfully discriminates both conditions, is the spectral slope, as indicated by a clear separation of the two states along the first dimension (on the x-axis).

When discerning wakefulness from REM sleep, information (as quantified by unique explained variance eta squared) about the underlying arousal state was significantly different between spectral slope, SO power and their interaction (Figure 3D). Post hoc permutation t-tests revealed that this effect was mostly driven by the spectral slope (slope: 0.29 ± 0.05 (mean ± SEM), SO: 0.05 ± 0.01; interaction: 0.06 ± 0.03; post-hoc permutation t-tests (Bonferroni corrected): Slope-SO: p < 0.001, obs. t_17_ = 4.29, d = 1.50, Slope-Int.: p < 0.001, obs. t_17_ = 4.84, d = 1.62; SO-Int.: p = 0.63, obs. t_17_ = 0.82, d = 0.29).

For NREM stage 3 sleep and wakefulness, there was no difference in explained variance between factors (Figure 3E; post-hoc permutation t-tests (Bonferroni corrected): Slope-SO: p = 0.406, obs. t_17_ = -1.11, d = -0.39; Slope-Int.: p = 0.422 obs. t_17_ = 1.09, d = 0.31; SO-Int.: p = 0.032, obs. t_17_ = 2.44, d = 0.88).

Between anesthesia and wakefulness, information about the state was again significantly different between factors (Figure 3F). As in the wake-REM differentiation, this could mostly be attributed to the spectral slope (post-hoc permutation t-tests (Bonferroni corrected): Slope-SO: p < 0.001, obs. t_8_ = 3.70, d = 1.75), Slope-Int.: p < 0.001, obs. t_8_ = 3.95, d = 1.87, SO-Int.: p = 0.019, obs. t_8_ = 2.67, d = 1.07).

The results from the GLM mirrored the findings from the LDA approach: When discerning wakefulness from REM sleep and anesthesia, the spectral slope was more informative about the underlying arousal state than SO power or the interaction of both factors. For NREM sleep stage 3, both slope and SO power contained a comparable amount of information. The interaction of both factors did not explain more unique variance than the sum of the univariate metrics, hence, indicating that the slope and SO power provide complimentary information about arousal states.

We carefully considered the reviewers’ proposal to replace the LDA approach but felt that the results as obtained from the LDA offer an additional advantage, namely a very intuitive comparison as quantified as percent correct to interrater reliability in sleep scoring (80.6% for R&K and 82% for AASM; (Danker-Hopfe et al., 2009). Therefore, we elected to present both approaches side-by-side in the revised manuscript:

“General Linear Model: When discerning wakefulness from REM sleep, information (as quantified by unique explained variance eta squared) about the underlying arousal state was significantly different between the spectral slope, SO power and their interaction (repeated-measures ANOVA permutation test: p < 0.001, F_1.16, 19.74_ = 19.69). […] The spectral slope enabled an improved classification and contained more unique information about arousal state compared to slow wave power when differentiating wakefulness from both propofol anesthesia and REM sleep and was comparable when discerning wakefulness from N3 sleep.”

Furthermore, the discussion of prediction accuracies should include a reference to the "ground truth" that is predicted here, the ratings of a trained specialist which are 80% reliable.

We thank the reviewers for the recommendation. This information can be found in the following paragraph:

This finding indicates that the spectral slope constitutes a marker that successfully discriminates REM sleep from wakefulness solely from the electrophysiological brain state. Critically, the accuracy of this classification is comparable to trained personnel, since the inter-rater reliability between sleep state scoring experts is typically about 80% (Danker-Hopfe et al., 2009).

3) One of the major claims of the paper is that the effect of interest is consistent across techniques (EEG, iEEG) and brain states (propofol, sleep). To support this claim, the authors would need to be consistent in their analysis pipelines of the different datasets.

We appreciate the reviewers’ input and therefore, carefully assessed the influence of different variables on spectral slope estimation. In particular, we identified the following parameters, which were queried:

Power - Method for power calculation:

To evaluate the impact of different spectral decompoisition methods for subsequent spectral slope estimation, we recalculated the power spectra using the original multi-taper approach, a single Hanning taper, a periodogram and Welch’s method (Figure 2—figure supplement 6). We calculated a signal-to-noise ratio (mean divided by standard deviation) and found that the multitaper approach provides significantly cleaner spectral estimates as compared to the other methods in all tested frequency bands (n = 20; cluster permutation test: p < 0.001).

More specifically:– Why were different length segments (and taper numbers) used for sleep vs anesthesia? Would be better for direct comparison to keep these parameters the same.

We appreciate the opportunity to clarify our motivation and our rationale for the chosen parameters and then present supporting data.

Motivation and rationale

While we agree with the reviewers that the same settings for all recordings would facilitate an easier direct comparison, we want to point out that sleep and anesthesia are different neurophysiological states. While they share some characteristics including a reduced arousal level, they also exhibit different temporal dynamics and spectral features. Specifically, the duration of these states typically differs – while a full night of sleep often lasts for 6 – 10 hours, surgical anesthesia is often only maintained for 1 – 3 hours or less. Here, we find similar recordings lengths of 7.99 ± 0.01 hours (mean ± SEM) for sleep and 3.81 ± 0.22 hours for anesthesia.

a) Segment Length:

Sleep scoring is typically carried out in 30 second segments under the assumption that neuronal activity stays relatively stationary within this time frame (Prerau et al., 2017). We therefore adapted a 30 second segment length for our slope estimation during sleep as the 30 second staging was the only “ground truth” available.

For anesthesia, no such convention exists. While many studies used 1 to 2 second segments (e.g. (Colombo et al., 2019; Gao et al., 2017), we decided to utilize a longer segment length of 10 seconds as this approach offered the opportunity to use a Multitaper approach for power estimation in the examined frequency range. A segment length of 30 seconds, on the other hand, resulted in pronounced temporal smoothing of slope dynamics under anesthesia compared to 10 second segments (Figure 1—figure supplement 3A).

b) Taper Numbers:

The number of the dpss tapers used in the Multitaper method is a direct result of the segment length and our chosen frequency smoothing of 1 Hz (Prerau et al., 2017):

THBP=SL*fR2 The time-half-bandwidth product (THBP) is a result of the segment length (SL) and the frequency resolution (fR). With this parameter we can now calculate the numbers of tapers (NT) used in the estimate:NT=(2*THBP)−1 For anesthesia, we chose a segment length of 10 seconds, and a frequency resolution of 1 Hz, resulting in a THBP of 5:10*12=5 This means for a segment length of 10 seconds and a frequency resolution of 1 Hz, we needed 9 dpss tapers for a clean estimate: (2*5)−1=9 Supporting data:

To further evaluate the influence of segment length and number of tapers on our findings, we recalculated the spectral slope estimates for both scalp EEG groups: For anesthesia, we repeated the analysis with 30 second segments (Figure 1—figure supplement 3) and for sleep with 10 second segments (Figure 2—figure supplement 7).

**Author response table 1. resptable1:** 

**Scalp EEG**	**original**	**new**
**Anesthesia** (n = 9)	10 second segments	30 second segments
Wakefulness	-1.84 ± 0.30	-1.85 ± 0.33
Anesthesia	-3.10 ± 0.20	-2.94 ± 0.18
**Sleep** (n = 20)	30 second segments	10 second segments
Baseline rest	-1.87 ± 0.18	-1.90± 0.19
NREM 3	-3.46 ± 0.16	-3.42 ± 0.16
REM	-4.73 ± 0.23	-4.72 ± 0.22

Spectral slope estimates with original and new settings under anesthesia with propofol and in sleep, both in scalp EEG (Study 1 and 3; mean ± SEM).

Both new analyses resulted in comparable spectral slope patterns (Author response table 1, Figure 1—figure supplement 3, Figure 2—figure supplement 7), where the original and new observations were strongly correlated with r’s of 1 or 0.98, respectively (Pearson: anesthesia orig.-new: r = 1, p < 0.0001, sleep orig.-new: r = 0.98, p < 0.0001). Under anesthesia, however, the 30 second segment resulted in pronounced temporal smoothing of slope dynamics compared to 10 second segments (Figure 1—figure supplement 3a). We therefore kept the original segment lengths in the main manuscript but added our results as supplemental figures to facilitate a direct comparison between segment lengths and number of tapers.

– The iEEG studies and the scalp EEG studies use a very different reference. Bipolar for iEEG and common average for the scalp EEG studies (the reference is not clearly specified for study 3 though, please clarify).

The reviewers are correct in pointing out that we used different reference schemes in our recordings: For the anesthesia EEG (Study 1) as well as the scalp EEG contacts in intracranial sleep (Study 4) we used a common average reference. The sleep scalp EEG study (Study 3) was already recorded with bilateral linked mastoids as reference.

We agree with the reviewers that the reference scheme for Study 3 was not stated clearly enough (it could only be found in the data collection paragraph of Material and Methods). We therefore added a new sentence to the data preprocessing section: Study 3 – Sleep scalp EEG: The EEG was referenced to bilateral linked mastoids (…)

For both the intracranial recordings under anesthesia and in sleep (Study 2 and 4), we chose a bipolar reference scheme as this is a) the reference scheme commonly used in intracranial recordings (Helfrich et al., 2019) and b) the most conservative approach. For scalp EEG, a variety of reference schemes are possible and there is no consensus on which scheme is used best. The motivation for the different reference schemes was the following:

Study 1 – Anesthesia scalp EEG: The recording environment in the operating room during surgical anesthesia will always be suboptimal in terms of environmental noise. To address this, we employed a common average reference to remove common instrument /movement noise and to improve signal quality.

Study 3 – Sleep Scalp EEG: This dataset was recorded in the quiet setup of a sleep laboratory and was already referenced to linked bilateral mastoids (which were not available in other scalp EEG datasets). However, we did repeat the slope estimation using a Laplacian reference. This method strengthened the observed relationship between the hypnogram and the spectral slope (Laplacian: p_Spearman_ < 0.001, p_MI_ < 0.0001; partial correlation: p_Spearman_ < 0.001; Figure 2—figure supplement 2 and Author response image 2).

**Author response image 2. sa2fig2:** The influence of the choice of reference on the relationship of hypnogram and spectral slope. Original observation with bilateral linked mastoids reference (left panels) versus Laplacian re-reference (right panels) in scalp EEG recordings during sleep (n = 20). Upper row of topoplots: Cluster permutation test of Spearman rank correlation between hypnogram and time-resolved slope: * p < 0.05. Lower rows of topoplot: Mutual Information between time-resolved slope and hypnogram. Statistics with random block swapping: * p < 0.05.

Study 4 – Sleep Intracranial, scalp EEG contacts: Patients with intracranial electrodes received additional scalp EEG contacts (Fz, Cz, C3, C4, Oz) to facilitate gold-standard sleep scoring. As the coverage was too sparse for e.g. a Laplacian reference, we utilized a common average reference.

Our results replicated the findings from the Study 3, revealing that bilateral linked mastoid reference, Laplacian reference and common average reference provide similar spectral slope estimations. This finding implies that our results were robust to changes in reference schemes. To underscore this observation, we repeated out slope estimation under different reference schemes in the following 3.3 response.

One would expect that the choice of reference would have a large impact on the slope, but this does not seem the case. For example, local dynamics captured by bipolar referencing are in general very different from global effects such as those observed in scalp EEG. In fact, the Laplacian reference does not change the results. Could the authors discuss in more detail the role of referencing in their interpretation of the neurophysiological basis of 1/f dynamics?

We thank the reviewers for this comment and appreciate the opportunity to evaluate the influence of a certain reference scheme on the estimation of the spectral slope. We therefore recalculated the spectral slope for the sleep scalp EEG (Study 3) after using different reference schemes:

The original: bilateral linked mastoids reference

Laplacian reference (already in the initial version of the manuscript; LP)

Common average reference (CAR)

Clinical bipolar, also often called “Double Banana” (DB)

We found that while the absolute slopes values vary slightly, the overall slope pattern with more negative values during sleep compared to wakefulness is remarkably similar (Figure 2—figure supplement 8B). The average across states was comparable between original and CAR (n = 14; paired t-tests (uncorrected): Orig.-CAR: p = 0.065, t_13_ = 2.02, d = 0.36) and original and DB reference (Paired t-tests (uncorrected): Orig.-DB: p = 0.0875, t_13_ = -1.85, d = -0.32) but different between original and LP reference (Orig.-LP: p = 0.005, t_13_ = -3.35, d = -0.64). Slope estimation between original and all other reference schemes were strongly correlated (Pearson: Orig. – CAR: r = 0.90, p < 0.001; Orig. – DB: r = 0.77, p < 0.001; Orig. – LP: r = 0.80, p < 0.001).

When calculating the Mutual Information (MI) between the time-resolved slope and the hypnogram at electrode Fz, there was no significant difference between original and CAR (n = 14; Paired t-tests (uncorrected): Orig.-CAR: p = 0.94, t_13_ = -0.08, d = -0.01) or original and LP reference (p = 0.109, t_13_ = 1.72, d = 0.33) but a significantly higher MI for original compared to DB reference (Paired t-tests (uncorrected): Orig.-DB: p = 0.007, t_13_ = 3.23, d = 0.62).

Taken together, the choice of reference did not affect the overall pattern of results: A more negative slope was observed during sleep as compared to wakefulness. Critically, slope values derived from different reference schemes were strongly correlated.

4) Furthermore, we had some concerns regarding anatomical claims.The fact that sleep stage-slope correlation seems to be dominated by medial PFC/temporal sites is interesting, though might be biased by the iEEG montage, as those are also the regions with the most electrodes due to epilepsy monitoring. There should be a statistical analysis that shows that those regions have a significantly higher proportion of "differentiating" electrodes than other brain regions. In addition to a sound statistical analysis, the authors should provide a table with the number of total electrodes and the number of electrodes showing a significant change in slope, per brain region. As it is, it's hard to define the regions of interest based on the brain plots shown.

We thank the reviewers for their suggestions. We want to point out that we did not select electrodes in regions of interest but used all available electrodes that were non-epileptic and located in gray matter. See Figure 1—figure supplement 1.

155 out of 352 SEEG (44.03 %) followed the observed scalp EEG pattern of a more negative slope in sleep compared to wakefulness (Figure 2C), significantly more than chance (33 % chance level; chi-squared test: Χ2 = 8.20, p = 0.0042).

We agree with the reviewers that there might be a selection bias in anatomical location of contacts in medial temporal lobe (MTL) regions as temporal epilepsy is the most common form of epilepsy. To access MTL regions, electrodes are inserted through lateral temporal cortex (LTC). To access orbitofrontal, cingulate and insular regions, electrodes are inserted through the lateral prefrontal cortex (PFC). Both implantation schemes lead to a paucity of recording sites in parietal and posterior regions. As the implantation of all the electrodes is purely due to clinical consideration, it will remain one limitation of this method.

We carefully revisited the anatomical localization of the depth electrodes and collected the information about subregions and state- dependent slope modulation in a table, as requested by the reviewers (Table 2). In Study 3 – Intracranial sleep, the majority of sEEG electrodes were located in PFC (132/352, 37.5 %), followed by LTC (79/352; 22.4%) and MTL (48/352; 13.6%).

As the visual impression suggested, medial PFC exhibits a significantly higher fraction of electrodes with sleep state- dependent slope modulation than lateral PFC (chi-squared test: Χ2 = 33.56, p < 0.000). The same was true for MTL compared to LTC (Χ2 = 33.12, p < 0.0001). When comparing mPFC and MTL, which both showed high levels of state- dependent slope modulation, or lPFC and LTC, that showed only little modulation, there was no significant regional difference (mPFC – MTL: Χ2 = 1.89, p = 0.169; lPFC – LTC: Χ2 = 0.19, p = 0.659). These comparisons underscore that the differences in chi squared test are not driven by electrode number.

For completeness, we also included a table (Table 1) with the subregions of electrodes in Study 2 – Intracranial anesthesia although almost all electrodes exhibited a state- dependent slope modulation:

We included the tables in the manuscript and edited the following paragraph:

“We observed the same pattern – a more negative spectral slope in N3 and REM sleep as compared to wakefulness – in 155 of 352 SEEG (44.03 %; *chi-squared test against chance-level (33%)*: *Χ2* = 8.20, p = 0.0042; Figure 2C, Figure 2—figure supplement 3). Importantly, this analysis revealed that medial prefrontal cortex (mPFC) and medial temporal lobe structures (MTL; details see Table 2, Figure 2—figure supplement 3) exhibit a significantly larger fraction of electrodes showing sleep state- dependent slope modulation compared to their lateral counterparts (*chi-squared tests:*mPFC – lateral PFC: *p < 0.0001, Χ2* = 33.56, MTL – lateral temporal cortex: p < 0.0001, *Χ2* = 33.12), hence, converging on the same brain regions known to be the most relevant for sleep-dependent memory consolidation (Dang-Vu et al., 2008; Helfrich et al., 2018; Mander et al., 2013; Murphy et al., 2009).”

Additionally, it would have been much more powerful if the anatomical claims could be corroborated by the EEG recordings. Not sure source localization is possible on the EEG data given low electrode numbers, but that would really have strengthened the story. In current form it remains a little unclear what the main sources of all these effects are.

We thank the reviewers for this suggestion and we source-localized the slope difference between wakefulness and sleep in scalp EEG (Study 3, n = 19; one patient had to be excluded due to insufficient wake trials).

Cortical sources of the sensor-level EEG data were reconstructed by using a LCMV (linearly constraint minimum variance) beamforming approach (Van Veen et al., 1997) to estimate the time series for every voxel on the grid. A standard T1 MRI template and a BEM (boundary element method) headmodel from the FieldTrip toolbox (Oostenveld et al., 2011) were used to construct a 3D template grid at 1cm spacing in standard Montreal Neurological Institute (MNI) space. Electrode location was derived from a standard 10/20 template from the FieldTrip toolbox (Oostenveld et al., 2011). Prior to source projection, sensor level data was common average referenced and epoched into 30 second segments. To minimize computational load, we selected a two second data segment from the center of every epoch to construct the covariance matrix. The LCMV spatial filter was then calculated using the covariance matrix of the sensor-level EEG data with 5% regularization. Spatial filters were constructed for each of the grid positions separately to maximally suppress activity from all other sources. The resulting time courses in source space then underwent spectral analysis using a Fourier transform after applying a Hanning window. The PSD slope of every segment was then estimated using linear regression in the range from 30-45 Hz as outlined before. To obtain a similar contrast to the intracranial analysis depicted in Figure 2C, mean slope values during sleep were subtracted from the mean slope during wakefulness at every voxel in source space and then source-interpolated onto a standard template brain in MNI space (Figure 2—figure supplement 3). Note that δ slope is reported as positive values whereas slope difference is depicted as a negative value. Both intracranial as well as the source-interpolated scalp EEG results revealed a focus in medial prefrontal areas (mPFC, Figure 2C – magenta, 2—figure supplement 3 – yellow).

Since it is challenging to localize sources to the MTL (in particular in the case of a 19 channel EEG), we employed an ROI-based approach focusing on PFC subregions. In line with our results from the intracranial sleep study (Study 4, Figure 2C and Figure 2—figure supplement 3), we defined the mPFC and dorsolateral PFC (dlPFC) as regions-of-interest (ROI; Figure 2—figure supplement 3). To identify ROI at source level, we used the Automated Anatomical Labeling (AAL) atlas as implemented in FieldTrip (Oostenveld et al., 2011). The ROI mPFC contained the following tissue labels: 23 & 24 – ‘Frontal_Sup_Medial_L & R’, 25 & 26 – ‘Frontal_Med_Orb_L & R’, 27 & 28 – ‘Rectus_L & R’, 31 & 32 – Cingulum_Ant_L & R’. The ROI dlPFC consisted of the following atlas tissue labels: 3 & 4 – ‘Frontal_Sup_L & R’, 7 & 8 – ‘Frontal_Mid_L & R’, 11 & 12 – ‘Frontal_Inf_Oper_L & R’, 13 & 14 – ‘Frontal_Inf_Tri_L & R’ and 15 & 16 ‘Frontal_Inf_Orb_L & R’.

In the mPFC ROI the slope decreased from wakefulness to sleep by 0.54 ± 0.23 (mean ± SEM) whereas in dlPFC it decreased by 0.36 ± 0.22. This slope difference between wakefulness and sleep was significantly different from zero for mPFC (paired t-test: p = 0.018) but not for dlPFC (p = 0.074). This difference in slope modulation between mPFC and dlPFC was significant (p = 0.036) in line with our observations in intracranial EEG.

Taken together, the observed anatomical pattern from the beamformer analysis of the sleep scalp EEG fits the previous results from the intracranial sleep study.

However, a beamformer analysis of only 19 channel EEG has only limited explanatory power and should be interpreted cautiously. We therefore refrained from adding the analysis to the main manuscript, especially since the manuscript already features intracranial recordings in humans. However, we introduced a new supplemental figure (Figure 2—figure supplement 3) depicting the results which is referred to in the following sentence of the result section:

“We established that the spectral slope closely tracks the hypnogram. […] Given the spatial heterogeneity of intracranial responses (Parvizi and Kastner, 2018), there was a remarkable convergence on medial PFC that resembled the pattern observed at source level (Figure 2—figure supplement 3) and the overlying scalp EEG electrode Fz (Figure 2).”

5) The fitting of 1/f spectral exponents should be improved and frequency ranges should be motivated more clearly.Although it is correct that Gao et al., 2017, used a frequency range of 30-50 Hz for their 1/f PSD fit, this choice was motivated by the linear decrease of power in LFP recordings within that frequency range.

We would like to clarify our motivation to choose the 30 – 45 Hz range for spectral slope estimation. Please note that we now also incorporated a detailed empirical validation in the manuscript.

There is a wide variation of slope estimations from different frequency ranges (Colombo et al., 2019; He et al., 2010; K. J. Miller et al., 2009b; Miskovic et al., 2019; Pereda et al., 1998; Shen et al., 2003). Very low frequencies below 1 Hz were out of reach due to limitations with the DC amplifier in our recordings (He et al., 2010). We also excluded frequencies above 45 Hz as we were careful not to include any filtering artifacts from the 50 Hz line noise in the European recordings. We then adapted this setting for all datasets across recording modalities for reasons of consistency.

The choice to calculate the slope estimate with this particular frequency band stemmed from the following considerations:

(Gao et al., 2017) used a 30 – 50 Hz frequency range in their paper with propofol anesthesia in monkeys; it therefore seemed reasonable to expect that this frequency range would also be suitable in human anesthesia data.

Prominent neuronal oscillations mostly occur below 30 Hz and could possibly confound the estimation of the 1/f slope estimate. Especially states like propofol anesthesia and NREM sleep stage 3 are characterized by strong oscillations (e.g. α at 8 – 12 Hz and δ at 1 – 4 Hz under propofol (Purdon et al., 2013) and slow waves (< 1.25 Hz) and spindles at12 – 16 Hz in sleep (Prerau et al., 2017). To circumvent this problem, we a) used a fit above the frequency range where most oscillations occur and b) validated the choice by a removal of the oscillatory component using irregular resampling (IRASA; (Wen and Liu, 2016).

Note the IRASA method is computationally expensive and time consuming. This would prevent any further use of the marker in a clinical on-line setting like monitoring depth of anesthesia. Hence, we decided to use the approach to fit above the oscillatory frequency domain to offer an easy and computationally effective tool that is not confounded by oscillations.

Additional appeal to use the 30-45 Hz frequency range came from the possible mechanistical explanation: (Gao et al., 2017) showed in the modelling part of their paper that the slope in this frequency range tracks changes in the ratio of inhibition to excitation. As both propofol anesthesia and NREM sleep stage 3 are associated with an increase in inhibition (Brown et al., 2011; Niethard et al., 2016), the slope in that range should reliably track changes from wakefulness to these states of reduced arousal.

We then validated the chosen frequency range in both the sleep study with scalp EEG (Study 3) and the sleep study with intracranial recordings (Study 4). While both sleep studies provided us with a hypnogram that could pose as a “ground truth” for the evaluation of different slope fits, the intracranial sleep data had the additional benefit of allowing slope fits above 45 Hz.

Center frequency for fit:

We evaluated the relationship between hypnogram and time-resolved slope as a function of different center frequencies (± 10 Hz around center frequency starting from 20 up to 150 Hz: 10-30 (20), 20-40 (30), 30-50 (40) and so on) and found that spectral slope estimates only correlated significantly/ had a significant Mutual Information (MI) with the hypnogram if center frequencies up to 40 Hz were selected for the fit Figure 2—figure supplement 5A

Length of fit:

We evaluated the relationship between hypnogram and time-resolved slope as a function of fit lengths (from 30 Hz onwards with a 10 Hz increase of fit length up to 100 Hz: 30-40, 30-50, 30-60 and so on). The results showed that spectral slope estimates could be fitted with variable fit length from 30 Hz onwards and still resulted in a significant correlation/ MI with the hypnogram, see Figure 2—figure supplement 5B

Fit to low frequencies:

We evaluated fits to lower frequencies starting from 1 to 5 Hz with an increasing length of additional 5 Hz per fit after discounting the oscillatory components from the power spectrum by means of irregular resampling (IRASA; Figure 2—figure supplement 5c). The rationale for using a method like IRASA before attempting a slope fit was the occurrence of strong low frequency oscillations like slow waves which could possibly distort the slope estimate. When comparing the MI between the spectral slope fits and the hypnogram to the MIs of a surrogate distribution derived from a random block swapping procedure and the hypnogram, the 30 to 45 frequency range resulted in significantly higher MI than the fits to lower frequencies, see Figure 2—figure supplement 5D.

The results of these calculations were part of the original version of the manuscript (old Figure S6; new Figure 2—figure supplement 5). However, we realize that the organization of the subpanels was non-optimal and led to some confusion about the connection between subpanels. We rearranged the results in the new version of the figure and hope that it is now easier to follow.

Taken together, our analyses regarding the frequency range settings for spectral slope estimation suggested that a center frequency up to 40 Hz with at least a 10 Hz range should be used if calculating the spectral slope for sleep EEG for 30 second segments. A spectral slope estimated from the 30-45 Hz range provided the highest MI with the hypnogram indicating that this frequency range is well suited to distinguish wakefulness from N3 and REM sleep.

However, oscillatory parts of the spectrum, whether of neural or non-neural origin (line noise), can aptly be fitted and excluded from a 1/f fit using the FOOOF package (https://fooof-tools.github.io/fooof/). The use of this or another software package (e.g. BOSC or eBOSC: https://github.com/jkosciessa/eBOSC) would allow the authors to directly asses the change of PSD exponents across different levels of arousal across wider frequency ranges without having to alter time-series first using IRASA.

We thank the reviewers for their suggestions. We would like to highlight that the IRASA method (Wen and Liu, 2016) was only used for one part of the analysis (fitting to frequency ranges below 30 Hz, compare 2c – Fit to lower frequencies). Therefore, all the analyses in the manuscript were performed on the recorded time-series data after artifact rejection, demeaning and detrending.

As suggested by the reviewer, we evaluated the influence of different slope fitting algorithms on spectral slope estimation and recalculated the slopes using both the FOOOF (Haller et al., 2018) and the eBOSC algorithm (Caplan et al., 2001; Kosciessa et al., 2020; Whitten et al., 2011) in Study 1 – Anesthesia scalp EEG (Figure 1—figure supplement 4) and Study 3 – Sleep scalp EEG (Figure 2—figure supplement 9) and added goodness of fit statistics.

Linear regression:

Both our approach as well as the eBOSC algorithm (Kosciessa et al., 2020) perform a linear regression on the PSD in log-log space, albeit with using either the polyfit.m function (in our case) or the robustfit.m in the case of eBOSC.

Model fit:

The FOOOF algorithm follows the following steps to estimate the oscillatory and aperiodic signal: First an exponential fit is performed to the power spectrum in semi-log space, then this fit is subtracted from the PSD. The residual signal is treated as a combination of oscillations and noise and is repeatedly fit with multiple gaussian fits to detect putative oscillatory peaks until the noise floor is reached. The detected peaks are then validated against the mixed oscillatory/noise signal and subtracted from the original PSD. The new residual signal is then fit again for a better estimate of the aperiodic signal. Both, the multi Gaussian fits and the improved aperiodic fit are then combined in a model (compare to Figure 3 (Haller et al., 2018)). When the PSD does not contain a bend, also called knee, in the aperiodic signal, then the algorithm is “equivalent to fitting a line in log-log space” (Haller et al., 2018).

**Author response image 3. sa2fig3:** The influence of different fit algorithms on spectral slope estimation in general anesthesia with propofol. a, 1/f spectral slope in wakefulness (Wake, red) and under anesthesia (Ana, blue) using the original polyfit (Orig., orange), FOOOF (lblue) or eBOSC (purple) algorithm (n = 9, averaged across electrodes). Paired t-tests (uncorrected): Orig.-FOOOF: p = 0.082, t8 = 1.99, d = 0.18; Orig.- eBOSC: p = 0.369, t8 = -0.95, d = -0.03, FOOOF - eBOSC: p = 0.118, t8 = -1.75, d = -0.21. n.s. – not significant. b, At Fz ,1/f slope from original polyfit (Orig., orange) and FOOOF (blue; left panel: r = 0.98, p < 0.001) and original and eBOSC (purple; right panel: r = 1.00, p < 0.001) are strongly correlated.c, Goodness of fit (R2) of the different slope estimates (Orig. - original polyfit (orange); FOOOF (light blue); eBOSC (purple)) to the power spectra in wakefulness and under propofol anesthesia at electrode Fz (n = 9). Permutation t-tests: Orig. - fooof: p = 0.538, Orig. - eBOSC: p = 0.726, fooof - eBOSC: p = 0.690. n.s. – not significant.

Study 1 – Anesthesia scalp EEG:

We evaluated all three slope fitting algorithms – polyfit, FOOOF and eBOSC – in both the 30 – 45 as well as the 1-40 Hz range (compare 2c – Fit to lower frequencies and Figure 1—figure supplement 4). Note that propofol anesthesia is characterized by strong oscillations, mainly in the δ (1-4) and α range (8-12 Hz) which might distort spectral slope estimates (Purdon et al., 2013).

30 – 45 Hz: When fitting the slope to the 30 – 45 Hz range, there was no significant difference between slope estimated derived from the original polyfit and eBOSC, FOOOF and eBOSC (Figure 1—figure supplement 4a/ Author response image 3A). The slope estimates derived from both FOOOF and eBOSC were strongly correlated with slopes values calculated by polyfit (Figure 1—figure supplement 4b/ Author response image 3B). Between fitting algorithms, there was no difference in effect size (Cohen’s d; Figure 1—figure supplement 4e) or goodness of fit to the power spectra (Author response image 3C).

1 – 40 Hz: When the spectral slope was calculated between 1 – 40 Hz, then the different slope fitting algorithms resulted in small but significantly different slope estimates (Figure 1—figure supplement 4D); possibly because some algorithms are more susceptible to oscillatory peaks in low frequency bands than others. Despite this difference, the slope values were still strongly correlated (Figure 1—figure supplement 4D). There was a significantly higher effect size for the Original compared to eBOSC (Orig._1-40_-eBOSC_1-40_: p = 0.004, obs. t_8_ = 3.42, d = 0.15) but not for the Original versus FOOOF (Orig._1-40_-fooof_1-40_: p = 0.092, obs. t_8_ = 1.38, d = 0.15) or FOOOF and eBOSC (fooof_1-40_- eBOSC_1-40_: p = 0.656, obs. t_8_ = 0.40, d = 0.04). When calculating the goodness of fit at electrode Fz, there was no difference between algorithms (Orig._1-40_-fooof_1-40_: p = 0.162, Orig._1-40_-eBOSC_1-40_: p = 0.705, fooof_1-40_- eBOSC_1-40_: p = 0.651).

Study 3 – Sleep scalp EEG:

When employing all three slope fitting algorithms, namely the polyfit, FOOOF and eBOSC, the resulting slope estimates were remarkanly similar although some small but consistent differences existed (n = 14; mean difference between slopes values: Orig.-FOOOF: 0.043 ± 0.022, Orig.-eBOSC: -0.003 ± 0.001). These differences were not significant between the original polyfit and the FOOOF algorithm but were between the original and eBOSC (Figure 2—figure supplement 9A). Both slopes derived from FOOOF and eBOSC were strongly correlated with slopes calculated by the original polyfit (Figure 2—figure supplement 9B, C). When calculating the goodness of fit to the PSD, there were no significant differences between the original and both FOOOF or eBOSC (Figure 2—figure supplement 9D).

Taken together, in both sleep and anesthesia, the choice of slope fitting algorithm did not change the observed pattern of more negative slop values during reduced arousal states. Although absolute slope values derived from the algorithms did slightly vary, they were strongly correlated with each other and their goodness of fits were comparable. Hence, the choice of slope fitting algorithm did not impact the ability of the spectral slope to differentiate between arousal states in sleep and anesthesia.

We added the following summary paragraph to the control analysis section in the revised manuscript, see subsection Evaluation of parameters for 1/f spectral slope estimation.

Moreover, we added the following paragraph to the discussion: subsection “Practical considerations for analyzing 1/f dynamics”

In fact, Colombo and colleagues show a steepening of the PSD between 1 and 40 Hz from awake rest compared to Propofol anesthesia, a result the authors could try to replicate and discuss using the proposed analyses techniques.

We appreciate the reviewers’ suggestion and recalculated the spectral slopes under propofol anesthesia using a frequency range from 1 – 40 Hz in line with Colombo et al., 2019 in Study 1 – Anesthesia scalp EEG (n = 9). When the slope was fitted to the 1-40 Hz frequency range, the absolute slope values changed slightly but the overall pattern of a more negative slope for anesthesia than for wakefulness persisted (Figure 1—figure supplement 4). To directly compare the fits to the 30-45 and the 1-40 Hz frequency range, we contrasted the effect sizes (Cohen’s d) of the wakefulness-anesthesia comparison of the two fits (Figure 1—figure supplement 4G) and found no significant difference between the two irrespective of fitting algorithm (Permutation t-test: Orig._30-45_ – Orig._1-40_: p = 0.773, obs. t_8_ = -0.84, d = -0.37; fooof_30-45_ – fooof_1-40_: p = 0.737, obs. t_8_ = -0.63, d = -0.28; eBOSC_30-45_ – eBOSC_1-40_: p = 0.672, obs. t_8_ = -0.47, d = -0.20).

Moreover, we recalculated how well both slope fits can differentiate between wakefulness and anesthesia with a Linear Discriminant Analysis (LDA). Both 1/f slope fits result in a higher percentage of correct classification compared to SO power (SO: 52.43 ± 1.04 % (mean ± SEM), slope_30-45_: 76.56 ± 3.56 %, slope_1-40_: 83.75 ± 2.31 %; permutation t-tests: Slope_30-45_ – SO: p < 0.001, observed (obs.) t_8_ = 6.10, d = 2.63; Slope_1-40_ – SO: p < 0.001, obs. t_8_ = 9.24, d = 3.88). When directly comparing the slope fits, the fit to 1-40 Hz performed better than the 30-45 Hz (p = 0.023, obs. t_8_ = 2.49, d = 0.83).

Taken together, while the absolute slope values varied, the overall slope pattern as well as the effect size between wakefulness and anesthesia was comparable between fits. However, the 1-40 Hz fit had a better wake-anesthesia classification performance. This performance difference might be due to an inclusion of both the low frequency (< 4 Hz) and the α frequency (8 – 12 Hz) range into the 1 – 40 Hz fit. Under propofol anesthesia, these frequency ranges are typically characterized by prominent oscillations (Purdon et al., 2013) that might distort spectral slope estimation leading to overlapping information between the two factors. Notably, this finding is contrast to our results in sleep, where an inclusion of low frequencies hampered the MI between the hypnogram (Figure 2 —figure supplement 5D). This implies that optimal fit ranges for spectral slope estimation could differ between arousal states.

In conclusion, both the 30-45 as well as the 1-40 frequency range seem suited to be used for spectral slope estimation under anesthesia in scalp EEG. However, the 30 – 45 Hz range had the highest MI between the hypnogram and the spectral slope in sleep in scalp EEG indicating that this specific range is well suited to be used in both sleep and anesthesia recordings.

Furthermore, goodness of fit statistics for the performed linear fits are missing and should be added for any kind of fitting procedure the authors decide to apply.6) Discussion could be more in depth in terms of:– connection with existing literature;

We thank the reviewers for their comments and carefully revisited the manuscript. The discussion now features a clear structure with several new paragraphs that explore the connection to the existing literature, the underlying physiology and the implications for understanding sleep physiology.

“Neurophysiological markers of arousal states:

Consciousness is commonly assessed on two axes – content (e.g. the experience) and level (e.g. vigilance; (Boly et al., 2013; Laureys, 2005). […]Here, we demonstrate that the non-oscillatory, aperiodic part of the power spectrum, which is devoid of prominent low-frequency oscillatory components and can be approximated by the 1/f decay of the power spectrum estimated from the 30 – 45 Hz frequency range, reliably differentiates wakefulness from all three states of reduced *arousal level*, namely REM, N3 sleep and general anesthesia with propofol.”

– underlying mechanisms/generation of the recorded signals;

“The neurophysiologic basis of 1/f dynamics:

1/f dynamics are observed across a variety of tasks (He et al., 2010; Kai J. Miller et al., 2009; K. J. Miller et al., 2009b; Voytek et al., 2015), change with lifespan (Voytek et al., 2015), and exhibit state-dependent variations during sleep (Freeman and Zhai, 2009; Leemburg et al., 2018; Miskovic et al., 2019; Robinson et al., 2011), and anesthesia (Colombo et al., 2019; Gao et al., 2017). […] Future studies involving single neuron recordings and optogenetic manipulation will be needed to unravel the precise relationship between population firing statistics and changes in the spectral slope. Comparative studies in rodents (Gao et al., 2017; Leemburg et al., 2018), non-human primates (Gao et al., 2017) and humans (Colombo et al., 2019; He et al., 2010; Kai J. Miller et al., 2009; K. J. Miller et al., 2009b; Miskovic et al., 2019) combined with modeling work (Chaudhuri et al., 2018; Robinson et al., 2011, 2001) has the potential to integrate the divergent findings into a coherent framework, which is critical to further elucidate the neurophysiologic basis of 1/f dynamics and their relationship to arousal levels.”

– interpretation of the findings and implications for our understanding of sleep vs wake processes;

“Functional significance of 1/f dynamics in arousal states:

Here, we found a decreased 1/f slope in N3 sleep, REM sleep and under general anesthesia compared to wakefulness in four independent studies. […] This notion of increased inhibition during REM sleep offers a likely mechanistic explanation for certain REM-defining phenomena, such as muscle atonia (Scammell et al., 2017) or the clinical observation that epileptic seizures during the night predominantly occur out of more excitable, highly synchronized NREM sleep and only rarely out of less excitable, desynchronized REM sleep (Ng and Pavlova, 2013).”

– actual discussion of practical use, i.e., how reliable is slope as a measure to distinguish these states on the single subject level.

“Importantly, while some overlap of absolute spectral slope values between rest and sleep existed when comparing across individuals (Figure 2—figure supplement 1A), we observed a consistent individual decrease of – 2.06 ± 0.21 (mean ± SEM) between rest and all sleep stages (Figure 2—figure supplement 1B; Rest-N1 = -1.95 ± 0.26, Rest-N2 = -1.81 ± 0.23, Rest-N3 = -1.59 ± 0.28, Rest-REM = -2.86 ± 0.25).”

[Editors' note: further revisions were suggested prior to acceptance, as described below.]

The issue we are still unclear on pertains to this claim:"Critically, the accuracy of this classification is comparable to trained personnel, since the inter-rater reliability between sleep state scoring experts is typically about 80%".However, if the ground truth here consists of the sleep staging done by trained experts, for whom we know the inter-rater reliability is typically about 80%, then a 100% accuracy of the classifier would be a 100% match with these expert raters (i.e., not necessarily a 100% accurate classification of sleep stage). In other words, the performance of the classifier is bound by the accuracy of the ground truth. Since we do not know the "true" sleep stage, we can only conclude that the classifier accurately predicts the experts' ratings 80% of the time here. Please add some language to make this explicit in the discussion of these results.

We agree with the reviewer’s comment and modified the following paragraph in the manuscript to clarify the accuracy of the classification:

“Note that classification performance is bound by the accuracy of the underlying sleep scoring as a ground truth. Since the inter-rater reliability between sleep scoring experts is typically about 80% (Danker-Hopfe et al., 2009), the classifier accurately predicts the experts' ratings in 80% of the time.”